# Accelerated plasma-cell differentiation in *Bach2*-deficient mouse B cells is caused by altered IRF4 functions

Kyoko Ochiai [ID][1][✉], Hiroki Shima[1], Toru Tamahara [ID][2], Nao Sugie[1], Ryo Funayama[3], Keiko Nakayama[3], Tomohiro Kurosaki[4,5] & Kazuhiko Igarashi [ID][1,6][✉]

## Abstract

Transcription factors BACH2 and IRF4 are both essential for antibody class-switch recombination (CSR) in activated B lymphocytes, while they oppositely regulate the differentiation of plasma cells (PCs). Here, we investigated how BACH2 and IRF4 interact during CSR and plasma-cell differentiation. We found that BACH2 organizes heterochromatin formation of target gene loci in mouse splenic B cells, including targets of IRF4 activation such as *Aicda*, an inducer of CSR, and *Prdm1*, a master plasma-cell regulator. Release of these gene loci from heterochromatin in response to B-cell receptor stimulation was coupled to AKT-mTOR pathway activation. In *Bach2*-deficient B cells, PC genes' activation depended on IRF4 protein accumulation, without an increase in *Irf4* mRNA. Mechanistically, a PU.1-IRF4 heterodimer in activated B cells promoted BACH2 function by inducing gene expression of *Bach2* and *Pten*, a negative regulator of AKT signaling. Elevated AKT activity in *Bach2*-deficient B cells resulted in IRF4 protein accumulation. Thus, BACH2 and IRF4 mutually modulate the activity of each other, and BACH2 inhibits PC differentiation by both the repression of PC genes and the restriction of IRF4 protein accumulation.

**Keywords** BACH2 Complex; Heterochromatin; *Bach2*-deficiency; IRF4 Protein Accumulation; The PU.1-IRF4 Heterodimer
**Subject Categories** Chromatin, Transcription & Genomics; Immunology

## Introduction

A transcription factor (TF) orchestrates a gene regulatory network (GRN) which is composed of its direct target genes and its upstream factors that regulate the activity of the TF (Singh, 2014).

Each TF recognizes a unique DNA sequence (i.e., DNA motif), and recruits chromatin regulators, such as chromatin remodelers and histone modifying enzymes, to specific genomic regions for gene regulation. In many cases, TF binds to composite DNA motifs which consist of different TF DNA motifs; a variety of partner TFs provides diversity in TF function in regulating GRNs. Some TFs also interfere with the binding of other TFs to a shared DNA motif by competition, or to an adjacent DNA motif by modulating chromatin accessibility or steric hindrance. Overall, kinds and combinations of TFs are distinct depending on cell types and stages of cell differentiation. The state of TFs in a cell is unique to a given cell and ultimately determines the fate of the cell by orchestrating multiple GRNs.

A characteristic feature of B cell lies in its ability to proliferate and to differentiate into antibody-producing plasma cell (PC) in response to antigen. PC differentiation is initiated by signaling from the antigen receptor (i.e., BCR; B-cell receptor), and the differentiation process is driven by various TFs which function collaboratively, competitively or independently. In particular, BLIMP-1 which is encoded by the *Prdm1* gene has been established as a master TF of PC differentiation (Shapiro-Shelef et al, 2003), and functions in both gene activation and repression (Minnich et al, 2016). The activation targets of BLIMP-1 include *Xbp1*, a regulator of the endoplasmic reticulum maturation for antibody production (Shaffer et al, 2004), and *Myc*, a critical TF for cell proliferation (Lin et al, 1997). Its repression target genes include *Pax5*, the master TF for B-cell identity (Lin et al, 2002), and *Aicda* encoding the cytidine deaminase AID. Prior to PC differentiation, activated B cells undergo somatic hypermutation (SHM) and class-switch recombination (CSR) of the antibody genes to acquire affinity maturation and functional diversification of antibodies in the germinal center (GC). AID introduces DNA damage at the immunoglobulin gene (Ig) locus for initiating the processes of SHM and CSR (Chaudhuri et al, 2003; Muramatsu et al, 2000). BLIMP-1 determines the cell fate to become PC by terminating the expression of B-cell-specific genes, including those for CSR and SHM. Therefore, the expression of *Prdm1* is strictly regulated

[1]Department of Biochemistry, Tohoku University Graduate School of Medicine, Seiryo-machi 2-1, Sendai 980-8575, Japan. [2]Division of Community Oral Health Science, Department of Community Medical Supports, Tohoku Medical Megabank Organization, Tohoku University, Seiryo-machi 2-1, Sendai 980-8573, Japan. [3]Division of Cell Proliferation, United Centers for Advanced Research and Translational Medicine, Tohoku University Graduate School of Medicine, Seiryo-machi 2-1, Sendai 980-8575, Japan. [4]Laboratory of Lymphocyte Differentiation, Immunology Frontier Research Center, Osaka University, Osaka 565-0871, Japan. [5]Laboratory for Lymphocyte Differentiation, RIKEN Center for Integrative Medical Sciences (IMS), Yokohama, Kanagawa 230-0045, Japan. [6]Center for Regulatory Epigenome and Diseases, Tohoku University Graduate School of Medicine, Seiryo-machi 2-1, Sendai 980-8575, Japan. ✉E-mail: kochiai@med.tohoku.ac.jp; ; igarashi@med.tohoku.ac.jp

during B-cell development and antigen-driven activation of B cells. In these sequential processes following antigen stimulation, TFs BACH2 (BTB and CNC homolog 2) and IRF4 (Interferon regulatory factor 4) cooperatively promote CSR (Ochiai and Igarashi, 2022) but oppositely regulate PC differentiation; while BACH2 represses the expression of *Prdm1* (Ochiai et al, 2006), IRF4 activates its expression (Sciammas et al, 2006). Therefore, these two TFs play a critical role in the determination of B-cell fate upon antigen stimulation.

BACH2 belongs to the BACH family, and recognizes the Maf recognition element (MARE), TPA response elements (TRE), and the cAMP response elements (CREs) (Reinke et al, 2013). Deficiency in *Bach2* severely reduces SHM and CSR in immunized mice, accompanied by elevated expression of *Prdm1* and *Xbp1* (Muto et al, 2004). BACH2 binds to the regulatory regions of the *Prdm1* gene (Ochiai et al, 2006; Ochiai et al, 2008) and recruits the complexes of the corepressors NCOR and histone deacetylase HDAC3 for gene repression (Tanaka et al, 2016). BACH2 protein amount remains high for a longer period in class-switching IgG-type B cells than in non-class-switching IgM-type B cells (Muto et al, 2010), suggesting that BACH2 determines whether B cells undergo CSR. Furthermore, BACH2 determines differentiation to memory B cells (Kometani et al, 2013). BACH2 is negatively regulated by the AKT-mTOR (mammalian target of rapamycin) kinase pathway, which reduces *Bach2* gene expression and promotes BACH2 protein degradation by mTORC1 (mammalian target of rapamycin complex 1)-mediated phosphorylation (Ando et al, 2016; Tamahara et al, 2017). Consistent with the fact that BCR stimulation is coupled with the AKT-mTOR pathway and initiates the dynamic alteration of chromatin architecture for initiating differentiation (Kieffer-Kwon et al, 2017), BACH2 is reduced in its function upon BCR activation (Tamahara et al, 2017). The alteration of chromatin architecture upon BCR stimulation may also reduce the expression of *Bach2*. Nonetheless, BACH2 needs to be maintained active and/or re-activated in B cells undergoing antibody CSR. The mechanism for the maintenance or reactivation of BACH2 under BCR stimulation remains an open question to be explored.

IRF4 has been identified as a partner of PU.1, an ETS family TF (Eisenbeis et al, 1995), which together binds to the ETS-IRF composite element (EICE) for B-cell gene activation (Brass et al, 1999). The PU.1-IRF4 heterodimer functions across B-cell development (Lu et al, 2003), and induces the expression of various genes including *Bcl6* and *Icosl* required for the GC reaction (De Silva and Klein, 2015; Liu et al, 2015; Ochiai et al, 2013; Ochiai et al, 2018). Furthermore, IRF4 alters its function by binding to additional DNA motifs in the process of PC differentiation. AP-1 is one of the IRF4 partners, and it recruits IRF4 to the AP-1-IRF composite element (AICE) for gene activation (Glasmacher et al, 2012). BATF, an AP-1 family TF, is transiently expressed in activated B cells, and forms DNA-binding complexes with IRF4 on the *Aicda* locus to induce its expression (Ochiai et al, 2018). Thus, the PU.1-IRF4 and BATF-IRF4 heterodimers cooperatively orchestrate GRNs promoting CSR prior to PC differentiation. Layered upon this regulation, IRF4 protein is accumulated toward PC differentiation and becomes to recognize the interferon-stimulated response element (ISRE), another DNA motif bound by the IRF4 homodimer (Ochiai et al, 2013). Accumulated IRF4 activates the expression of *Prdm1* by binding to an ISRE within the intron (Sciammas et al, 2006). It is suggested that the induction of *Irf4* by BLIMP-1 forms the IRF4-BLIMP-1-positive feedback loop, which

drives and fixates PC differentiation (Minnich et al, 2016). Due to the modulation of multiple GRNs via diverse DNA motifs, IRF4 is indispensable for both CSR and PC differentiation (Klein et al, 2006; Sciammas et al, 2006). In a previous study, BACH binding motifs were detected adjacent to IRF4 binding motifs in activated B cells (Xu et al, 2015), suggesting a competitive or collaborating roles of BACH2 and IRF4. However, their functional relationship has never been explored.

The AKT-mTOR pathway has important roles in modulating immunoglobulin gene recombination during B-cell development (Clark et al, 2014; Omori et al, 2006). AKT binds to cellular membrane lipids phosphatidylinositol 4,5-bisphosphate (PI(4,5)P$_2$) or phosphatidylinositol (3,4,5) triphosphate (PIP$_3$) produced by phosphoinositide 3-kinase (PI3K), which is activated via signaling from receptors including BCR. AKT activation is modulated by its phosphorylation, and, in particularly, phosphorylation at serine 473 (Ser473) brings about full activation of AKT. AKT promotes mTORC1 activity by phosphorylating components of mTORC1, followed by phosphorylation of p70S6K (p70 ribosomal protein S6 kinase), which in turn phosphorylates mTOR at serine 2448 (Ser2448) (Chiang and Abraham, 2005; Holz and Blenis, 2005; Rosner et al, 2010). Since the AKT-mTOR pathway inhibits CSR by modulating the function of TFs, including BACH2 (Omori et al, 2006; Tamahara et al, 2017), its activity needs to be restricted in B cells undergoing CSR. It remains unclear how the AKT-mTOR pathway is kept in check during B-cell activation.

Here, we tried to examine how the roles of BACH2 and IRF4 were integrated into the regulation of CSR and PC differentiation. We found the regulatory mechanism of AKT as a key switch of BACH2 and IRF4. In addition to inhibiting the BACH2 activity, AKT increased IRF4 protein, which led to elevated *Prdm1* expression induced by the IRF4 homodimer and resulted in PC differentiation. On the other hand, AKT activity was reduced by the PU.1-IRF4 heterodimer. Surprisingly, in B cells from *Bach2*-deficient mice, PU.1-IRF4 function was decreased, and IRF4 promoted PC differentiation without committing the IRF4-BLIMP-1-positive feedback loop.

## Results

### BACH2 represses gene loci for CSR and plasma cell in an H3K9me3-mediated manner

In previous reports, we have set up an in vitro PC differentiation system using resting B (naive B) cells purified from B1-8 mice, which allowed us to analyze both B-cell activation and PC differentiation in response to BCR signaling (Ochiai et al, 2021) (Fig. 1A). Considering that BACH2 is regulated via the BCR-AKT-mTOR pathway (Ando et al, 2016; Tamahara et al, 2017), we examined how BACH2 function was regulated under BCR signaling using this system. We found that a majority of BACH2 protein was localized around nuclei in naive B cells (Tamahara et al, 2017), and BACH2 was co-localized with the nuclear membrane protein LAMIN B1 in both naive B and activated B cells (Fig. 1B). These observations raised the possibility that the gene repression controlled by BACH2 involved the association of target gene loci at the nuclear membrane. To explore this possibility, endogenous BACH2 was purified using anti-BACH2 antibodies from B1-8[hi]

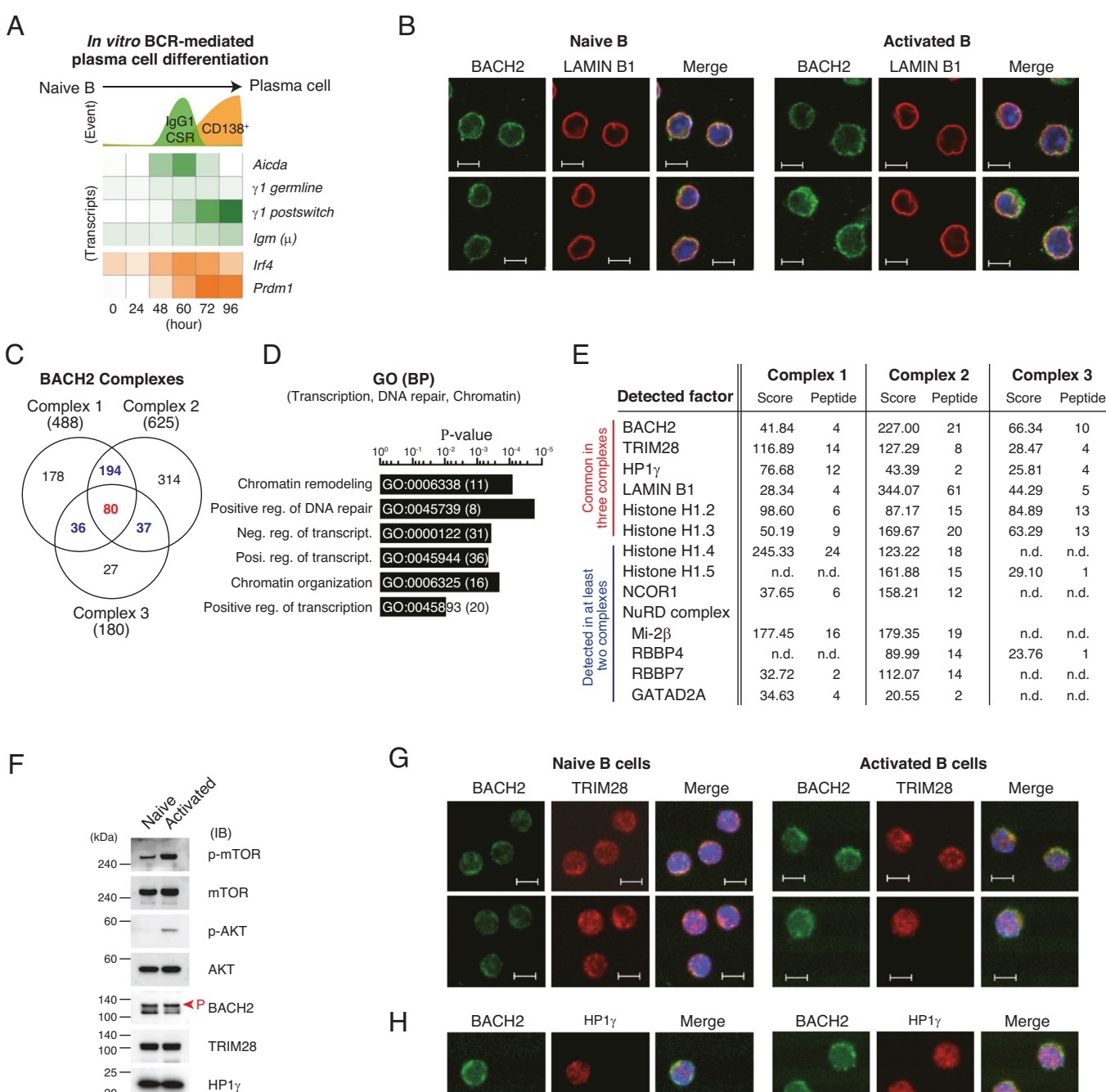

A

**In vitro BCR-mediated plasma cell differentiation**

B

**Naive B**    BACH2    LAMIN B1    Merge

**Activated B**    BACH2    LAMIN B1    Merge

C

**BACH2 Complexes**

D

**GO (BP)**
(Transcription, DNA repair, Chromatin)

| | P-value |
|---|---|
| Chromatin remodeling | GO:0006338 (11) |
| Positive reg. of DNA repair | GO:0045739 (8) |
| Neg. reg. of transcript. | GO:0000122 (31) |
| Posi. reg. of transcript. | GO:0045944 (36) |
| Chromatin organization | GO:0006325 (16) |
| Positive reg. of transcription | GO:0045893 (20) |

E

| | Detected factor | Complex 1 | | Complex 2 | | Complex 3 | |
|---|---|---|---|---|---|---|---|
| | | Score | Peptide | Score | Peptide | Score | Peptide |
| Common in three complexes | BACH2 | 41.84 | 4 | 227.00 | 21 | 66.34 | 10 |
| | TRIM28 | 116.89 | 14 | 127.29 | 8 | 28.47 | 4 |
| | HP1γ | 76.68 | 12 | 43.39 | 2 | 25.81 | 4 |
| | LAMIN B1 | 28.34 | 4 | 344.07 | 61 | 44.29 | 5 |
| | Histone H1.2 | 98.60 | 6 | 87.17 | 15 | 84.89 | 13 |
| | Histone H1.3 | 50.19 | 9 | 169.67 | 20 | 63.29 | 13 |
| Detected in at least two complexes | Histone H1.4 | 245.33 | 24 | 123.22 | 18 | n.d. | n.d. |
| | Histone H1.5 | n.d. | n.d. | 161.88 | 15 | 29.10 | 1 |
| | NCOR1 | 37.65 | 6 | 158.21 | 12 | n.d. | n.d. |
| | NuRD complex | | | | | | |
| | Mi-2β | 177.45 | 16 | 179.35 | 19 | n.d. | n.d. |
| | RBBP4 | n.d. | n.d. | 89.99 | 14 | 23.76 | 1 |
| | RBBP7 | 32.72 | 2 | 112.07 | 14 | n.d. | n.d. |
| | GATAD2A | 34.63 | 4 | 20.55 | 2 | n.d. | n.d. |

F

(IB)
p-mTOR
mTOR
p-AKT
AKT
P BACH2
TRIM28
HP1γ
αTUBULIN

G

**Naive B cells**    BACH2    TRIM28    Merge

**Activated B cells**    BACH2    TRIM28    Merge

H

BACH2    HP1γ    Merge    BACH2    HP1γ    Merge

mice B cells at 12 h after BCR stimulation, and interacting proteins were identified using mass spectrometry. Then, the components of BACH2 complex were determined as specific detection with anti-BACH2 antibodies, or more than two-fold protein score with anti-BACH2 antibodies than control IgG. Furthermore, independently purified three BACH2 complexes were compared, and 80 factors were commonly detected in three BACH2 complexes (Fig. 1C).

These factors were enriched with GO terms related to "chromatin remodeling", "DNA repair" and "transcripts" (Fig. 1D; Appendix Table S1). LAMIN B1 was detected as one of these core 80 factors. The known BACH2 interactors NCOR1 and NuRD complexes were detected in two of the three samples (Fig. 1E), confirming that they function with BACH2 in primary B cells. Furthermore, heterochromatin-related factors, including TRIM28 and HP1γ

**Figure 1. BACH2 interacts with H3K9me3-binding proteins in mouse splenic B cells.**

(A) Schematic view of differentiation events and transcripts using in vitro BCR-mediated PC differentiation. Splenic B cells purified from B1-8$^{hi}$ mice were stimulated with IL-2, IL-4, IL-5, CD40L, and NP-ficol. (B) Immunohistochemistry of BACH2 (green) and nuclear membrane protein LAMIN B1 (red) in naive B and activated B cells. (C) Venn diagram comparing numbers of the BACH2 complex carry out components. Three independent BACH2 purifications were performed using B1-8$^{hi}$ splenic B cells stimulated for 12 h, and examined the interacting components using LC-MS/MS. Each contained 488 or 625 or 180 components, respectively. Red, 80 proteins commonly included in all the complexes; Blue, the numbers of proteins shared by two complexes. (D) A gene ontology (GO) analysis of the BACH2 interacting proteins detected in at least two complexes. The *P*-value of enriched biological process (BP) related to transcription, DNA repair or chromatin are shown. The unique GO numbers and the number of proteins belonging to each GO are shown in bars. (E) Selected BACH2 interacting proteins were indicated with the protein scores and numbers of peptide detected in each BACH2 complex. (F) Immunoblot analyses of the indicated proteins. Red arrowhead indicates phosphorylated BACH2. αTUBULIN was used as an internal control. kDa kilodalton. Immunohistochemistry of BACH2 with TRIM28 (G) or HP1γ (H) in naive B and activated B cells. (B, F, G, H) Naive B cells, without stimulation; activated B cells, stimulated for 24 h. Data information: blue indicate nuclei stained with Hoechst 33342, and scale bars are 5 µm for all immunohistochemistry images. Source data are available online for this figure.

which recognize histone H3 lysine 9 tri-methylation (H3K9me3), and linker histone H1 were consistently identified in the BACH2 samples. Consistent with our previous report that BACH2 is phosphorylated by the AKT-mTOR pathway and localized in the cytoplasm (Ando et al, 2016), BCR stimulation increased the phosphorylation form of BACH2 (p-BACH2) and resulted in BACH2 accumulation in the cytoplasmic region (Fig. 1B,F). While TRIM28 and HP1γ were unchanged at the protein level between naive B cells and activated B cells (Fig. 1F), they co-localized with BACH2 in both naive B cells and activated B cells (Fig. 1G,H). These observations suggest that the BACH2-mediated target gene repression involves H3K9me3.

It was reported that the number of chromatin regions with enriched H3K9me3 became less in activated B cells than in naive B cells (Kieffer-Kwon et al, 2017). They utilized LPS, IL-4, and anti-CD180 antibodies for B-cell activation. In a previous report, we have reported that BACH2 is required for CSR in activated B cells stimulated with LPS and IL-4 (Muto et al, 2010). Therefore, their H3K9me3 ChIP-seq data are supposed to contain BACH2 regulatory regions losing the modification upon B-cell activation. In their ChIP-seq data, 143,396 and 88,825 H3K9me3 peaks in total were detected in naive B cells and activated B cells, respectively (Fig. 2A). To investigate the possible relationship of BACH2-mediated target gene repression with H3K9me3, we extracted genes regulated by BACH2 and H3K9me3 in naive B cells. From their H3K9me3 ChIP-seq data in naive B and activated B cells (Kieffer-Kwon et al, 2017), 25,677 peaks were detected in both naive B cells and activated B cells, while 116,778 or 62,382 peaks were uniquely detected in either of the B cells. Particularly, we focused on the 116,778 peaks, corresponding to 18,972 genes, which presumably lost H3K9me3 modification upon B-cell activation. As a cohort of H3K9me3-regulated genes in naive B cells, we also used 618 genes repressed by non-chromatin protein PC4, which regulates H3K9me3-mediated heterochromatin formation of non-B cells genes in naive B cells (Ochiai et al, 2020).

To identify BACH2 target genes from those genes, we utilized two data sets of BACH2 ChIP-seq. One was our previous data obtained in *Ebf1*-deficient pre-pro-B cells (Itoh-Nakadai et al, 2017). A total of 12,128 peaks, corresponding to 8790 genes, were extracted with enriched in BACH motif (Fig. 2B; Dataset EV1; Appendix Fig. S1). The other dataset was newly obtained from activated B1-8$^{hi}$ B cells. Compared with pre-pro-B cells, much fewer 664 peaks, corresponding to 627 genes, were identified (Fig. 2B; Dataset EV1). Importantly, sequences obtained from BACH2 ChIP-seq in activated B cells were also enriched in BACH motif with the significant *P*-value, and more than half, 55.50%, of targets

contained the BACH motif (Appendix Fig. S1). These results suggested that BACH2 ChIP-seq in activated B cells effectively exhibits BACH2-binding genomic regions. Importantly, many of the genes related to CSR or plasma cell showed BACH2 binding in pre-pro-B cells but not in activated B cells. On the other hand, genes related to hematopoietic progenitors, non-B lineage immune cells or early B cells showed BACH2 binding in both pre-pro-B cells and activated B cells, and these genes included *Kit*, *Ly96*, *Cish*, and *Cxcr4* (Dataset EV1). These observations indicated that BACH2 represses the expression of non-B-cell genes in both early B cells and activated B cells, and only a selected set of genes were de-repressed upon B-cell activation. Therefore, for further analysis, we applied genes bound by BACH2 in pre-pro-B cells but not in activated B cells as a loss of BACH2 binding in activated B cells.

To extract genes that lost both BACH2 binding and H3K9me3 modification upon B-cell activation, four gene sets, lost H3K9me3 modification in activated B cells, upregulated in *Sub1*-deficient B cells, BACH2 ChIP-seq in pre-pro-B cells and activated B cells were compared (Fig. 2B). Among the genes which lost H3K9me3 modification upon B-cell activation, 184 genes were commonly regulated by BACH2 and PC4, and 256 genes were regulated by PC4 but not by BACH2. Importantly, 6957 genes were extracted as genes that lost both BACH2 binding and H3K9me3 modification (Fig. 2B; Dataset EV1). These genes included known BACH2 target genes related to plasma-cell differentiation, such as *Prdm1* and *Ccnd3* (Ochiai et al, 2006; Tamahara et al, 2017). Thus, the extracted 6957 genes contained genes released from BACH2-mediated heterochromatin upon B-cell activation. Transcription start site (TSS) around these genes were reduced for H3K9me3 modification and increased for H3K27ac modification in activated B cells (Fig. 2C), confirming that these genes were activated from naive B to activated B cells. To confirm this interpretation, the expression of these genes was examined using our previous transcriptome data along PC differentiation (Ochiai et al, 2018). Two sets of gene clusters were extracted from upregulated genes (Fig. 2B, bottom). Cluster 1 included 456 genes (887 BACH2 peaks) which were transiently upregulated at 60 h, whereas cluster 2 included 831 genes (1770 BACH2 peaks) which were gradually upregulated toward PCs (Dataset EV2). BACH binding motif was detected as de novo top motif in sequences of BACH2-binding regions of the genes in both clusters (Fig. 2D,G). Genes in cluster 1 were enriched in "mRNA processing", "cell cycle", "transcription", "isotype switching", "SHM and DNA repair" in GO analysis (Fig. 2E,F; Dataset EV3). *Aicda*, an essential enzyme for CSR and SHM (Chaudhuri et al, 2003; Muramatsu et al, 2000), and *Batf*, an inducer of *Aicda* (Ochiai et al, 2018), as well as DNA repair factors

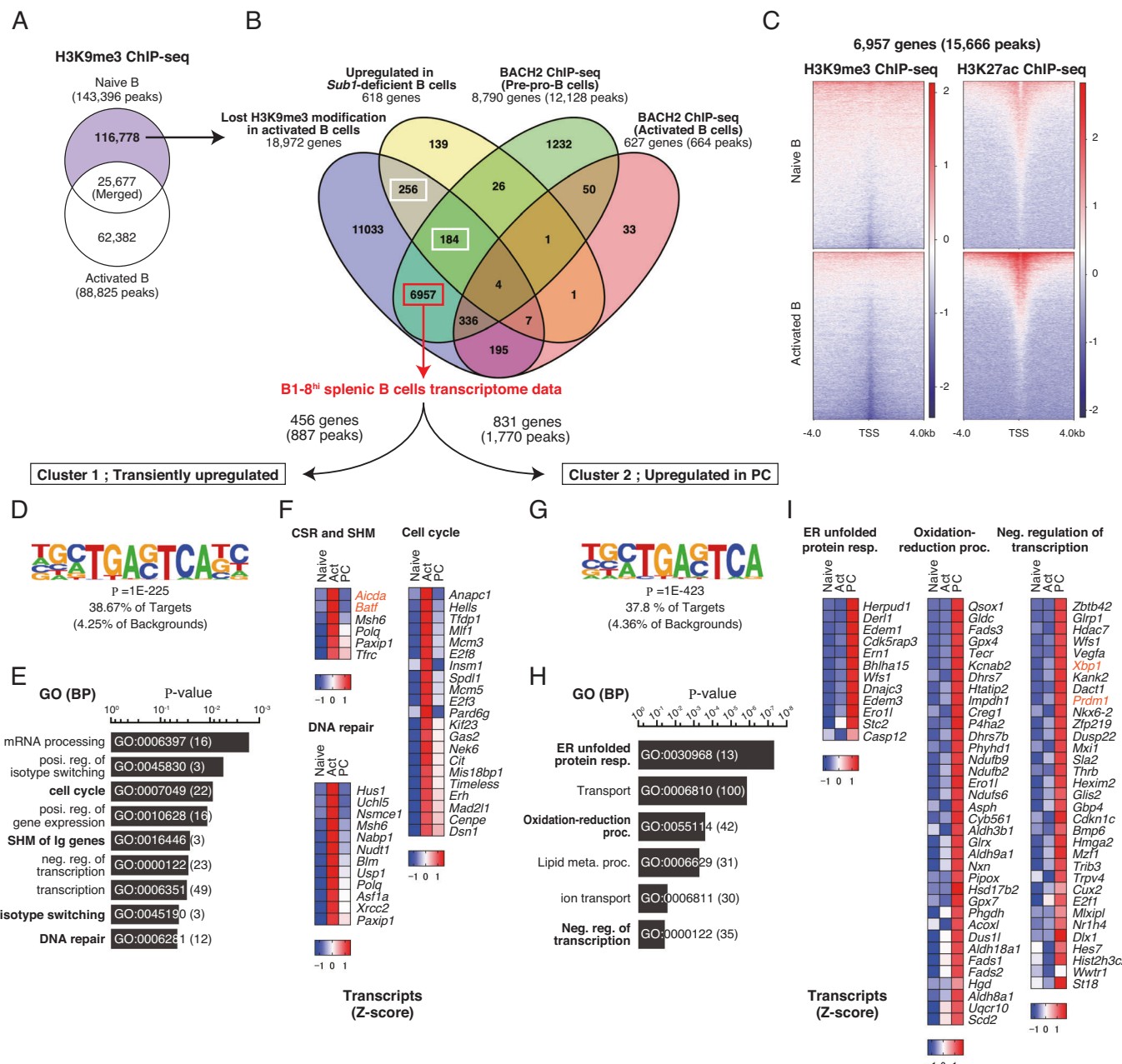

**Figure 2. BACH2-bound genomic regions lost H3K9me3 modification upon B-cell activation.**

(**A**) Extraction of H3K9me3 modification in naive and activated B cells. H3K9me3 ChIP-seq data in resting B cells and activated B cells were obtained from GSE82144. Activate B cells; activated with LPS (50 µg/mL), IL-4 (5 ng/mL) and anti-CD180 (0.5 µg/mL) for 24 h. (**B**) Extraction of the genes regulated by BACH2- and H3K9me3 in naive B cells. Venn diagram comparing BACH2 ChIP-seq in *Ebf1*-deficient pre-pro-B cells (GSE87503; 8290 genes, 12,128 peaks), BACH2 ChIP-seq in B1-8hi splenic B cells activated for 1 h (627 genes, 664 peaks), H3K9me3 modification lost in activated B cells (18,972 genes, 116,778 peaks), and upregulated in *Sub1*-deficient B cells (GSE145952; 618 genes). Among genes which lost H3K9me3 modification in activated B cells, 184 genes were shared between BACH2 and PC4, and 6957 genes were uniquely regulated by BACH2. (**C**) The H3K9me3 and H3K27ac distribution in the 6957 genes (15,666 peaks). H3K9me3 ChIP-seq and H3K27ac ChIP-seq, GSE82144. (**D–I**) Clustering of 6957 genes using transcriptome data. Cluster 1: transiently upregulated, 456 genes corresponding to 887 BACH2-binding peaks (**D–F**). Cluster 2: upregulated in plasma cell (PC), 831 genes corresponding to 1770 BACH2-binding peaks (**G–I**). (**D, G**) De novo motif identified using HOMER, shown with enrichment *P*-value, % of targets and backgrounds. (**E, H**) Selected GO terms in BP, shown with a modified Fisher Extract *P*-value for gene-enrichment analysis. (**F, I**) Transcripts of genes from indicated GO terms, shown with Z-score. Act, activated B cells at 60 h after stimulation; PC, CD138+ cells at 96 h.

were included in this cluster (Fig. 2F). Genes in cluster 2 were enriched in various cell functions such as "ER", "transport", "oxidation", "lipid metabolism" and "transcription" (Fig. 2H,I; Dataset EV3). *Prdm1* and *Xbp1*, well-known PC regulators, were

included in this cluster (Fig. 2I). These results indicate that BACH2 organizes heterochromatin-mediated silencing of genes for both CSR and PC in naive B cells. BACH2 inactivation may be necessary for initiating the activation of CSR and PC genes.

## Post-transcriptional accumulation of IRF4 protein in *Bach2*−/− mice B cells

In our previous reports, we showed the accelerated PC differentiation, with skipping CSR and the upregulation of *Prdm1* expression, in *Bach2*-deficient mice B cells (Muto et al, 2010; Muto et al, 2004; Ochiai et al, 2006). We next generated *Bach2*-deficient mice with B1-8^hi background (hereafter *Bach2*−/− mice), and performed transcriptome analysis comparing B cells purified from wild-type (*Bach2*+/+) and *Bach2*−/− mice, and examined transcripts of BACH2 target genes (Fig. 3A). The transcripts of *Aicda* and *Batf*, transiently upregulated BACH2 target genes (Fig. 2F), were not increased, and that of *Aicda* was rather decreased in *Bach2*−/− B cells. The transcripts of *Prdm1* and *Xbp1*, BACH2 target genes upregulated in

PCs (Fig. 2I), were significantly upregulated in *Bach2*−/− B cells. These results confirmed that *Bach2*−/− B cells were already committed to PCs.

Since *Prdm1* is induced by IRF4, we speculated that IRF4 is upregulated in *Bach2*−/− B cells. However, *Irf4* gene transcripts were reduced in *Bach2*−/− B cells (Fig. 3A). The enrichment of H3K27ac, a histone modification for gene activation, was reduced at the *Irf4* locus in *Bach2*−/− B cells, along with much fewer transcripts than *Bach2*+/+ naive B cells and PCs (Fig. 3B). Despite the reduced *Irf4* transcript level, IRF4 protein was rather increased in *Bach2*−/− B cells (Fig. 3C; Appendix Fig. S2). These results were further validated using RT-qPCR. Consistent with our previous reports (Muto et al, 2010; Muto et al, 2004), the expression of *Prdm1* was consistently much higher in *Bach2*−/− B cells than *Bach2*+/+ B cells

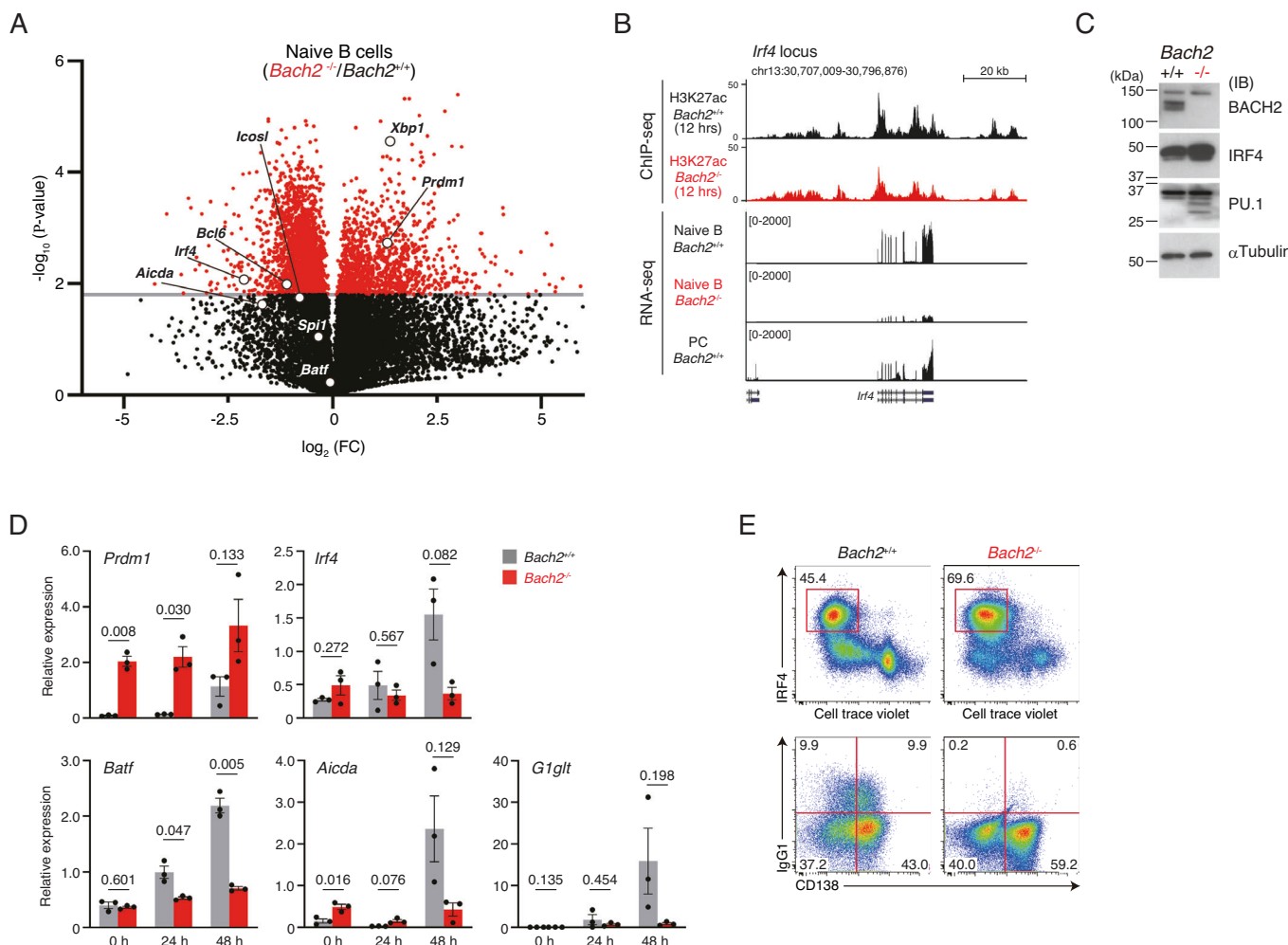

**Figure 3.  IRF4 protein accumulation by post-transcriptional mechanism in *Bach2*−/− B cells.**

Comparison analyses of splenic B cells from B1-8^hi:*Bach2*+/+ (hereafter, *Bach2*+/+) mice and B1-8^hi:*Bach2*−/− (hereafter, *Bach2*−/−) mice. (A) Volcano plots for RNA-seq data. *Y* axis, *P*-value ($-\log_{10}$); *x* axis, fold change (FC) ($\log_2$); gray line, the threshold of 1.8 for *P*-value by *t*-test. (B) H3K27ac enrichment and transcripts at the *Irf4* locus. H3K27ac ChIP-seq, splenic B cells stimulated for 12 h. PC, CD138+ cells at 96 h. (C) Immunoblot analyses of BACH2, IRF4, and PU.1. αTUBULIN was used as an internal control. kDa kilodalton. (D) Quantification of the *Prdm1, Irf4, Batf, Aicda*, and germline γ1 transcripts (*G1glt*) at the indicated time. (E) Intracellular IRF4 with cell division (upper), and surface IgG1 and CD138 (lower) at day 4. (B, D, E) Splenic B cells were stimulated with IL-2, IL-4, IL-5, CD40L, and NP-ficol. Data information: (B) H3K27ac ChIP-seq are representative of one or duplicate using *Bach2*+/+ or *Bach2*−/− B cells. RNA-seq are representative of three independent samples for each. (D) Data show the average values ± SD (error bars) acquired from one experiment using three mice for each genotype. *P*-value by *t*-test using R. (E) Data are representative of three independent experiments. Source data are available online for this figure.

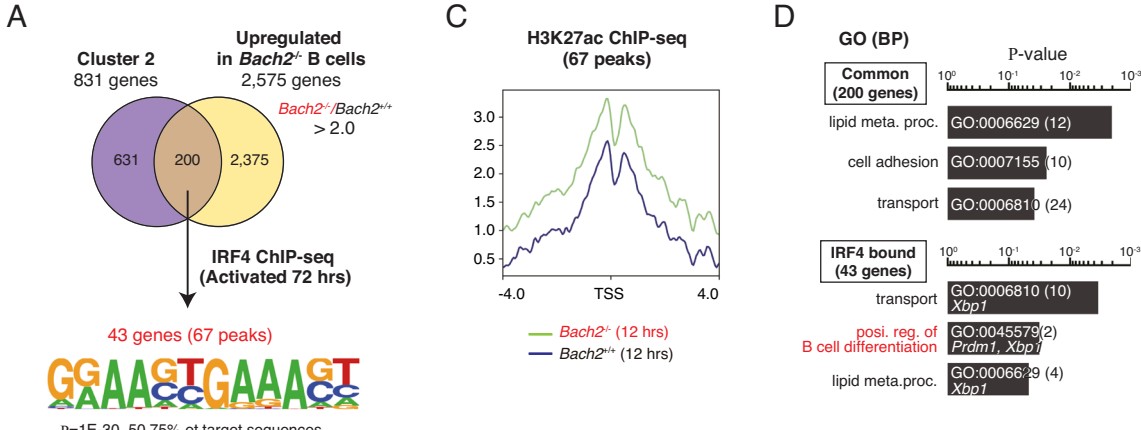

**Figure 4. IRF4 governs PC differentiation in *Bach2⁻/⁻* B cells.**

(A) Extraction of the IRF4 binding targets from upregulated genes in *Bach2⁻/⁻* B cells. Upregulated 2575 genes, *Bach2⁻/⁻/Bach2⁺/⁺* > 2.0 in the transcriptome analysis in Fig. 3A, were co-analyzed with cluster 2 genes in Fig. 2, which were bound by BACH2 and lost H3K9me3 modification upon activation and induced in PCs, and 200 genes were extracted. These genes were analyzed for IRF4 binding, and 43 genes, corresponding to 67 peaks, were extracted. Sequences within these IRF4-bound regions were analyzed for enriched motif using HOMER, shown with enrichment *P*-value and percentage of target sequences. ISRE, interferon-stimulated response element. (B) Transcripts of IRF4-bound 43 genes in *Bach2⁺/⁺* B cell, *Bach2⁻/⁻* B cell and *Bach2⁺/⁺* plasma cell (PC), shown with log₂ expression. (C) The H3K27ac distribution at TSS of 43 genes (67 peaks) in *Bach2⁺/⁺* and *Bach2⁻/⁻* B cells. (D) GO analysis of common 200 genes (upper) and IRF4-bound 43 genes (lower) shown with a modified Fisher Extract *P*-value for gene-enrichment analysis. BP biological process. (E–G) Presentation of BACH2 and IRF4 binding, H3K27ac enrichment, and transcripts. (E) The *Prdm1* locus (F) the *Tnfrsf17* locus and (G) the *Xbp1* locus. Red arrows, the regions containing indicated ISRE motif. BACH2 ChIP-seq, *Ebf1⁻/⁻* pre-pro-B cells (GSE87503). IRF4 ChIP-seq, *Bach2⁺/⁺* B cells stimulated for 72 h. H3K27ac ChIP-seq, *Bach2⁺/⁺* B cells stimulated for 12 h. RNA-seq data, shown in Fig. 2. PC, CD138⁺ cells at 96 h. (H) ChIP-qPCR of IRF4 binding and H3K27ac enrichment at the *Prdm1* (cns9, a red arrow in (E)). cns1, used as a negative control region at the *Prdm1* locus. Data show the average values ± SD (error bars) of three independent experiments. *P*-value by *t*-test using R.

(Fig. 3D). On the other hand, while the expression of *Irf4*, *Aicda* and *G1* germline transcript (*G1glt*) was induced along the time course in *Bach2⁺/⁺* B cells, these were not induced in *Bach2⁻/⁻* B cells. Regardless of the low *Irf4* expression, IRF4 protein was accumulated in a large population of activated *Bach2⁻/⁻* B cells, which differentiated into PC with higher frequency than *Bach2⁺/⁺* B cells without undergoing CSR (Fig. 3E; Appendix Fig. S2). Therefore, IRF4 protein was accumulated in *Bach2⁻/⁻* B cells independently of its mRNA expression, suggesting that the IRF4-BLIMP-1-positive feedback loop is not operating in *Bach2⁻/⁻* B cells. Furthermore, IRF4 failed to induce the expression of *Aicda* in this situation. The reason is presumably no induction of *Batf*, which recruits IRF4 to AICEs at the *Aicda* locus for its induction.

## Accumulated IRF4 governs the PC-GRN in *Bach2⁻/⁻* B cells

To confirm that IRF4 protein accumulation in *Bach2⁻/⁻* B cells activated the GRN for PC, we examined upregulated genes in *Bach2⁻/⁻* B cells for IRF4 binding. We focused on the 831 BACH2 target genes which were upregulated in PCs (Fig. 2B, cluster 2). Among them, 200 genes were upregulated in *Bach2⁻/⁻* B cells (Fig. 4A). These genes were examined for IRF4 binding using IRF4 ChIP-seq at 72 h after activation, and 43 genes with 67 peaks were extracted including *Prdm1* and *Xbp1* (Fig. 4A,B; Dataset EV4). The sequences of these IRF4-bound peaks contained ISRE bound by the IRF4 homodimer, which can be formed upon IRF4 accumulation. The status of H3K27ac was increased around the TSS of these genes in *Bach2⁻/⁻* B cells (Fig. 4C). In GO term, the 200 genes upregulated in both *Bach2⁻/⁻* B cells and PCs were enriched with factors related to "lipid metabolism process", "cell adhesion" and "transport" (Fig. 4D). The 43 IRF4 target genes were enriched with terms related to "transport", "B-cell differentiation" and "lipid metabolic process". Transcripts of IRF4-targeted 43 genes were upregulated in both *Bach2⁻/⁻* B cells and PCs (Fig. 4B). Among these genes, both *Prdm1* and *Tnfrsf17*, which encodes B-cell maturation antigen BCMA, showed increase in H3K27ac in *Bach2⁻/⁻* B cells compared with *Bach2⁺/⁺* B cells (Fig. 4E,F). In contrast, the *Xbp1* locus did not show such an increase in H3K27ac (Fig. 4G). Their transcripts were increased in *Bach2⁻/⁻* B cells. These observations are consistent with the previous reports showing that, while *Xbp1* is induced by alternative splicing, *Prdm1* is induced at the transcription level upon PC differentiation (Muto et al, 2010; Ochiai et al, 2006). ChIP-qPCR analysis confirmed that the IRF4 binding and H3K27ac enrichment at *Prdm1* cns9 were increased in *Bach2⁻/⁻* B cells (Fig. 4H). Thus, accumulated IRF4 in *Bach2⁻/⁻* B cells activates PC-related genes via ISREs. It should be noted that these

genes were directly bound by BACH2 at separate sites from IRF4 binding regions (Fig. 4E–G) and repressed in B cells (Fig. 2B). Taken together, BACH2 organizes heterochromatin formation of these gene loci, and the loss of BACH2 presumably increases the accessibility of IRF4 to its regulatory regions.

## IRF4 failed to activate EICE-regulated genes in *Bach2⁻/⁻* B cells

Prior to inducing the PC-related genes, IRF4 promotes B-cell activation and CSR by binding to EICE and AICE, respectively. Its target genes such as *Bcl6* and *Icosl* promote antibody maturation in GC B cells (De Silva and Klein, 2015; Huang et al, 2014; Liu et al, 2015), and they are induced by EICEs bound by PU.1-IRF4 heterodimer (Ochiai et al, 2018; Ochiai et al, 2013). However, the transcripts of *Bcl6* and *Icosl* were downregulated in *Bach2⁻/⁻* B cells (Fig. 3A). These observations raised the possibility that EICE-regulated genes, including *Bcl6* and *Icosl*, were downregulated in *Bach2⁻/⁻* B cells. To clarify the possibility, we examined whether the downregulated genes in *Bach2⁻/⁻* B cells contain PU.1-IRF4 regulated genes. From 4216 genes downregulated in *Bach2⁻/⁻* B cells, 1353 genes, corresponding to 2602 peaks, were extracted as IRF4-bound genes using IRF4 ChIP-seq at 24 h after activation (Fig. 5A). Then, 969 genes, corresponding to 1582 peaks, were further extracted as PU.1-IRF4 co-bound genes using PU.1 ChIP-seq. The sequences bound by PU.1-IRF4 contained the EICE motif with a significant *P*-value (Fig. 5A; Dataset EV4). The status of H3K27ac was decreased in *Bach2⁻/⁻* B cells around the TSS of these genes (Fig. 5B), confirming their decreased transcriptional activity. In GO term, these genes were enriched in "transcription", "protein phosphorylation" and "dephosphorylation", "chromatin modification", "apoptosis" and "signal transduction" (Fig. 5C). *Bcl6* and *Icosl* were included in these genes, and their transcripts were reduced in both *Bach2⁻/⁻* B cells and PCs (Fig. 5D–F). Thus, PU.1-IRF4 target genes, regulated by EICEs, were downregulated in *Bach2⁻/⁻* B cells.

During the course of CSR, AKT activity needs to be kept low, and several phosphatases are known to inhibit the AKT pathway. PTEN (phosphatase and tensin homolog) inhibits the AKT pathway by dephosphorylating PIP₃ (Maehama and Dixon, 1998) and modulates the AKT pathway in GCB cells (Luo et al, 2019). PHLPP1 (PH domain leucine-rich repeat protein phosphatase) directly dephosphorylates AKT at Ser473 (Gao et al, 2005). Importantly, we found that *Pten* and *Phlpp1* were also regulated by PU.1-IRF4 and downregulated in *Bach2⁻/⁻* B cells (Fig. 5D). These transcripts were reduced, and the enrichment of H3K27ac was decreased at their gene loci in *Bach2⁻/⁻* B cells (Fig. 5G,H). To clarify the reason for the downregulation of EICE-regulated genes,

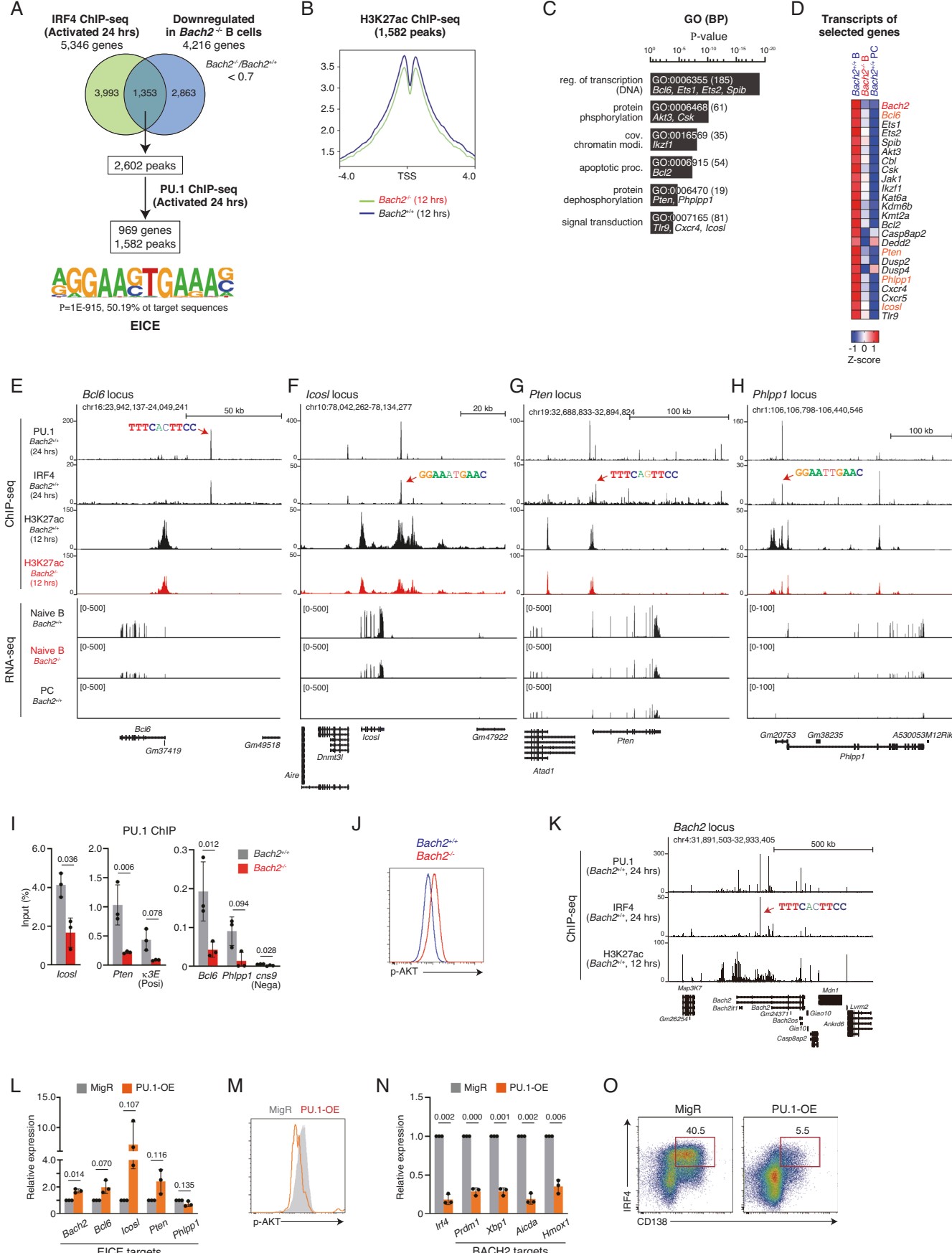

◄

**Figure 5. IRF4 failed to activate PU.1-IRF4 target genes in *Bach2⁻ᐟ⁻* B cells.**

(A) Extraction of PU.1-IRF4 binding targets from downregulated genes in *Bach2⁻ᐟ⁻* B cells. Downregulated 4216 genes, *Bach2⁻ᐟ⁻/Bach2⁺ᐟ⁺* < 0.7 in RNA-seq analysis, were co-analyzed with IRF4-bound 5393 genes, and 1353 genes, corresponding to 2602 peaks, were extracted. These peaks were further examined for PU.1 binding using HOMER mergePeaks, and 1582 peaks (969 genes) were extracted as PU.1-IRF4 co-bound regions. Sequences within these peaks were analyzed for an enriched motif using HOMER, shown with enrichment *P*-value and percentage of target sequences. EICE, Ets-IRF composite element. (B) The H3K27ac distribution at TSS of the 969 genes (1582 peaks) in *Bach2⁺ᐟ⁺* and *Bach2⁻ᐟ⁻* B cells. (C) GO analysis of the 969 genes shown with a modified Fisher Extract *P*-value for gene-enrichment analysis. BP biological process. (D) Transcripts of indicated 24 genes, selected from 969 genes, in *Bach2⁺ᐟ⁺* B cell, *Bach2⁺ᐟ⁺* PC, and *Bach2⁻ᐟ⁻* B cell, shown with Z-score. (E–H) Presentation of PU.1 and IRF4 binding, H3K27ac enrichment, and transcripts. (E) The *Bcl6* locus, (F) the *Icosl* locus, (G) the *Pten* locus, and (H) the *Phlpp1* locus. Red arrows, the regions containing indicated EICE motif. (I, J) Analysis of *Bach2⁺ᐟ⁺* and *Bach2⁻ᐟ⁻* naive B cells. (I) ChIP-qPCR of PU.1 binding at indicated genomic regions containing EICEs. *Prdm1* cns9, used as a negative control region. κ3E, possessing EICE and used as a positive control region. (J) Flow cytometry analysis of intracellular phospho-AKT at S473 (p-AKT). (K) Presentation of PU.1 and IRF4 binding at the *Bach2* locus. IRF4 ChIP-seq, *Bach2⁺ᐟ⁺* B cells stimulated for 24 h. PU.1 ChIP-seq, B1-8i splenic B cells stimulated for 24 h (GSE46607). H3K27ac ChIP-seq, *Bach2⁺ᐟ⁺* or *Bach2⁻ᐟ⁻* B cells stimulated for 12 h. RNA-seq data, shown in Fig. 2. PC, CD138⁺ cells at 96 h. (L–O) Analysis of PU.1 transduced (PU.1-OE cells). (L) The expression of EICE-target genes, *Bach2*, *Bcl6*, *Icosl*, *Pten* and *Phlpp1*. (M) Flow cytometry analysis of p-AKT. (N) The expression of *Irf4* and BACH2 target genes, *Prdm1*, *Xbp1*, *Aicda* and *Hmox1*. (O) Frequencies of intracellular IRF4 and surface CD138. Data information: (I, L, N) data show the average values ± SD (error bars) of three independent experiments. *P*-value by *t*-test using R. (J, M, O) Data are representative of three mice for each genotype or transduced cells. Source data are available online for this figure.

we examined the efficiency of PU.1 binding to EICE-contained regions. While PU.1 bound to the regulatory regions at the *Icosl*, *Pten*, *Bcl6*, and *Phlpp1* loci as well as the Ig κ3'E region, a positive control region of EICE (Eisenbeis et al, 1995), in *Bach2⁺ᐟ⁺* B cells, PU.1 binding to these regions was reduced in *Bach2⁻ᐟ⁻* B cells (Fig. 5I). Nuclear localization of PU.1 was similarly observed in both *Bach2⁺ᐟ⁺* and *Bach2⁻ᐟ⁻* B cells (Appendix Fig. S3). The expression of *Spi1*, encoding PU.1, did not alter (see Fig. 3A). Thus, PU.1 was dissociated from EICEs, resulting in the failure to activate the EICE-regulated genes in *Bach2⁻ᐟ⁻* B cells.

## PU.1 re-activates BACH2 function in activated B cells

Accompanied by the downregulation of *Pten* and *Phlpp1*, the level of phosphorylated AKT at Ser473 (p-AKT), the activation form of AKT, was elevated in *Bach2⁻ᐟ⁻* B cells (Fig. 5J). The AKT pathway is one of the important cell signaling pathways in the determination of CSR and PC differentiation (Omori et al, 2006), and the negative regulation of BACH2 is a part of the AKT function (Ando et al, 2016; Tamahara et al, 2017). Consistently with a previous study implicating in the regulation of *Bach2* by PU.1 via EICE (Wang et al, 2019), we detected PU.1-IRF4 binding to the *Bach2* locus in activated B cells (Fig. 5K). From these observations, we speculated that PU.1, cooperatively with IRF4, maintains BACH2 by inducing its transcription and inhibiting AKT activity via the induction of *Pten* and/or *Phlpp1* in activated B cells. To examine this possibility, PU.1 was transduced in activated B cells. The expression of *Bach2*, *Bcl6*, *Icosl*, and *Pten* were induced in PU.1 transduced (PU.1-OE) cells, while that of *Phlpp1* was not induced (Fig. 5L). The level of p-AKT was reduced in these cells (Fig. 5M), suggesting the effect of induced *Pten* expression. Importantly, the expression of BACH2 targets genes, *Prdm1*, *Xbp1*, *Aicda*, and *Hmox1*, as well as *Irf4* was dramatically reduced in PU.1-OE cells (Fig. 5N). The frequency of IRF4ʰⁱCD138⁺ PCs was predominantly reduced in these cells (Fig. 5O). Therefore, we conclude that PU.1 promotes the BACH2 function, and the mechanism involves the induction of *Bach2* expression and inhibition of AKT activity.

## Increased AKT activity promotes IRF4 protein accumulation in *Bach2⁻ᐟ⁻* B cells

AKT activation facilitates PC differentiation (Omori et al, 2006). Contrary to PU.1 transduction, transduction of constitutively active

AKT (CA-AKT) enhanced the percentage of IRF4ʰⁱCD138⁺ population (Appendix Fig. S4). Therefore, we examined the relevance of AKT activity to IRF4 protein accumulation. Upon B-cell activation, the level of p-AKT was elevated in CD138⁺ PCs compared to CD138⁻ cells regardless of IgG1 positivity and hence CSR (Fig. 6A). When activated B cells were sorted depending on IRF4 protein amount, IRF4ʰⁱ and IRF4ˡᵒ populations were enriched in CD138⁺ PCs or CD138⁻IgG⁺ cells, respectively, and the level of p-AKT was much higher in the IRF4ʰⁱ population than the IRF4ˡᵒ population (Fig. 6B). Next, we examined the effects of AKT inhibition on IRF4 accumulation. AZD5363, a selective inhibitor of AKT-mediated phosphorylation, was supplemented in cell culture medium at 48 h after activation, and the effect was analyzed at 96 h (Fig. 6C). The increased amounts of AZD5363 reduced the level of phosphorylated mTOR at Ser2448 (p-mTOR), which is promoted at the downstream of AKT signaling, without altering the ratio between p-AKT and total AKT (Fig. 6D). The frequency of CD138⁺ PCs and IRF4ʰⁱ population was also decreased with the increased amount of AZD5363 (Fig. 6E). Thus, AKT activation promotes IRF4 accumulation in activated B cells.

To examine whether IRF4 protein is accumulated by the elevated AKT activity in *Bach2⁻ᐟ⁻* B cells, splenic B cells were incubated in the culture medium supplemented with AZD5363. BACH2 is phosphorylated by mTORC1, and p-BACH2 was reduced with AZD5363 treatment in *Bach2⁺ᐟ⁺* B cells (Fig. 6F). IRF4 protein was not affected in these cells. In contrast, it was higher in *Bach2⁻ᐟ⁻* B cells and was reduced with the AZD5363 treatment. Therefore, IRF4 protein was accumulated by the elevated AKT activity in *Bach2⁻ᐟ⁻* B cells. We surmise that the downregulation of *Pten* and *Phlpp1* contributed to the increase of AKT activity, resulting in IRF4 protein accumulation in *Bach2⁻ᐟ⁻* B cells.

## Altered PU.1-IRF4 function and IRF4 regulation in *Bach2⁻ᐟ⁻* mice follicular B cells

Taken together, we found that *Bach2⁻ᐟ⁻* B cells were accelerated for PC differentiation by two altered functions of IRF4. One was a reduction in the PU.1-IRF4 function, which resulted in the decreased expression of EICE-regulated genes, including *Bcl6*, *Icosl*, *Pten*, and *Phlpp1*. The other is IRF4 protein accumulation, which facilitated the expression of ISRE-regulated genes including *Prdm1*, *Xbp1*, and *Tnfrsf17*. In the secondary lymphoid organ, B cells are localized at marginal zone (MZ) or follicular (FO), and these cells are distinguished with cell surface

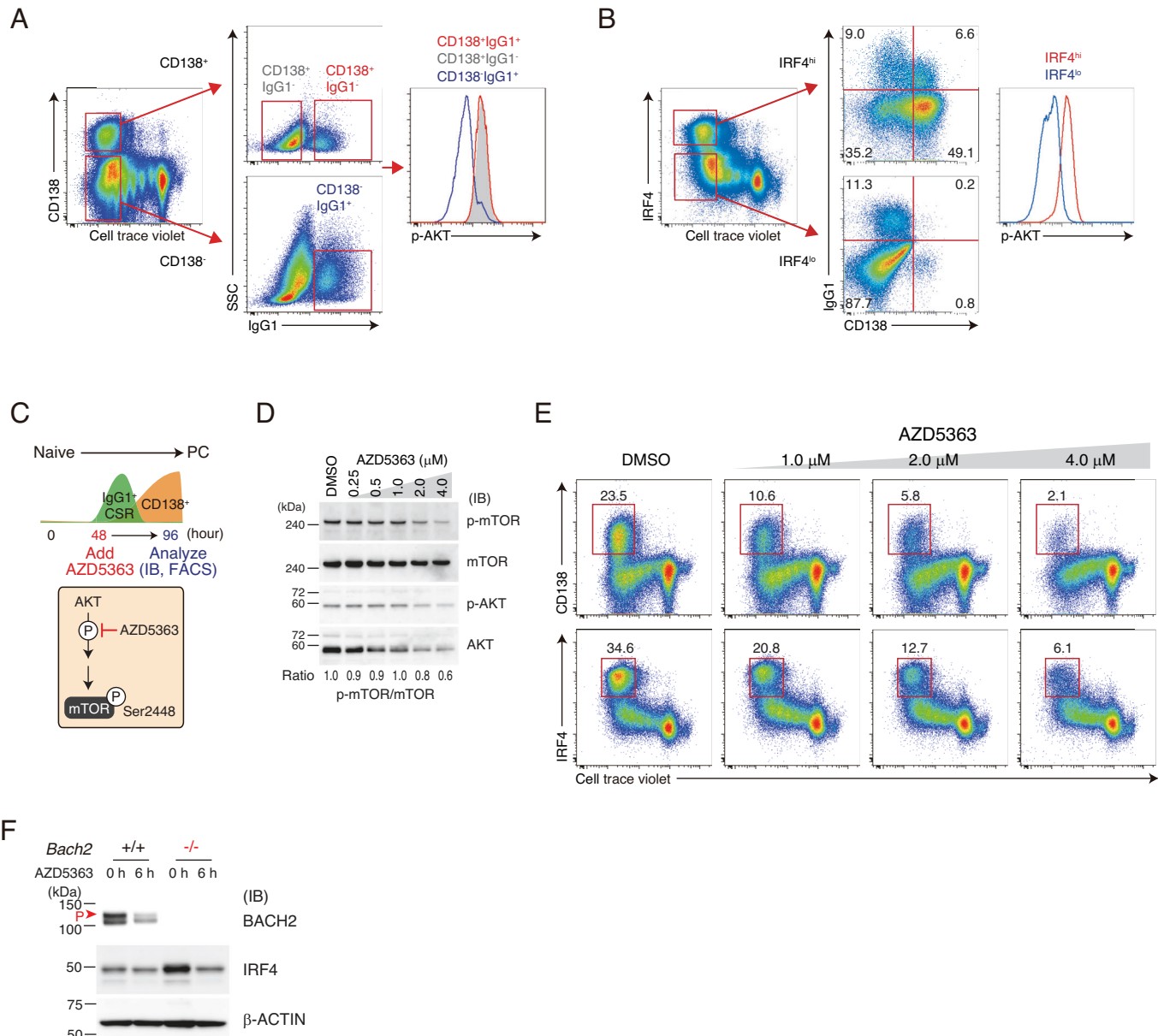

**Figure 6. Elevated AKT activation caused IRF4 protein accumulation in *Bach2*⁻/⁻ B cells.**

(A, B) Flow cytometry analysis of intracellular p-AKT in *Bach2*⁺/⁺ B cells activated at day 4. (A) CD138⁺ or CD138⁻ cells were sorted with IgG⁺ or IgG⁻ for comparison of p-AKT levels. (B) IRF4^hi or IRF4^low cells were examined for CD138 and IgG1 frequencies (middle) or p-AKT level (right). (C–E) Analysis of IRF4 protein level in activated B cells treated with an AKT inhibitor AZD5363. (C) Schematics of experimental flow. *Bach2*⁺/⁺ B cells were stimulated for 48 h, and cultured an additional 48 h by adding AZD5363 of indicated concentration in culture medium. (D) Immunoblot of total mTOR, p-mTOR, total AKT and p-AKT. Ratio, p-mTOR/mTOR (IB bands). p-mTOR, shown as an AKT substrate. kDa kilodalton. (E) Frequencies of surface CD138 (upper) and intracellular IRF4 (lower) with cell division. DMSO, used as a vehicle. Data are representative of two independent experiments. (F) Immunoblot analysis of BACH2 and IRF4 with AZD5363 treatment. *Bach2*⁺/⁺ or *Bach2*⁻/⁻ B cells were treated with 4.0 µM AZD5363 for 6 h, in the absence of cytokines and NP-ficol. Red arrowhead indicates phosphorylated BACH2. β-ACTIN was used as an internal control. kDa kilodalton. Data are representative of three independent experiments. Source data are available online for this figure.

markers (Fig. 7A). While the majority of MZ B cells differentiate to IgM-producing PCs in early time points, FO B cells undergo CSR and SHM to produce high-affinity antibodies. BACH2 was predominantly expressed in FO B cells (Fig. 7B) (Huang et al, 2014; Shinnakasu et al, 2016), and *Bach2*⁻/⁻ mice showed decreased FO B cells and increased MZ B cells (Fig. 7A). These observations suggest that *Bach2*-deficiency influenced FO B cell differentiation.

To examine whether the two altered functions of IRF4 pertain to FO B cells, we examined the expression of EICE- or ISRE-regulated genes in FO B and MZ B cells. The expression of EICE-regulated genes, *Bcl6, Icosl, Pten*, and *Phlpp1*, was decreased in *Bach2*⁻/⁻ FO B cells but not altered in *Bach2*⁻/⁻ MZ B cells (Fig. 7B), indicating that a reduction in the PU.1-IRF4 function was mainly observed in FO B cells. Importantly, the expression of *Prdm1*, an

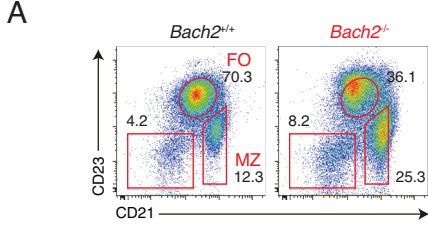

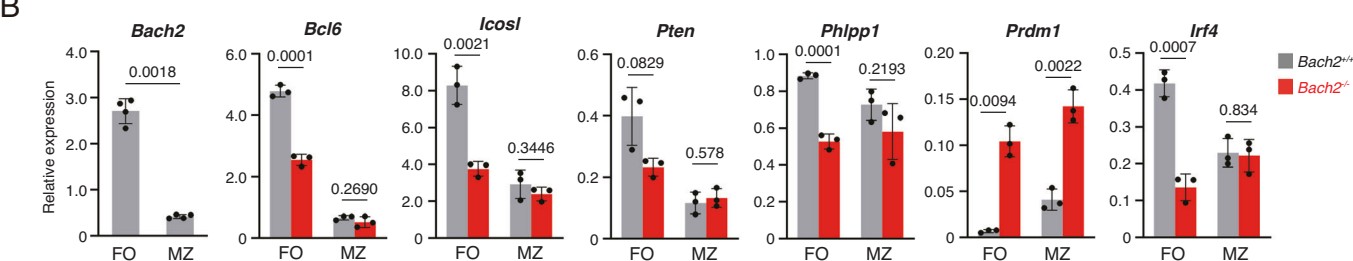

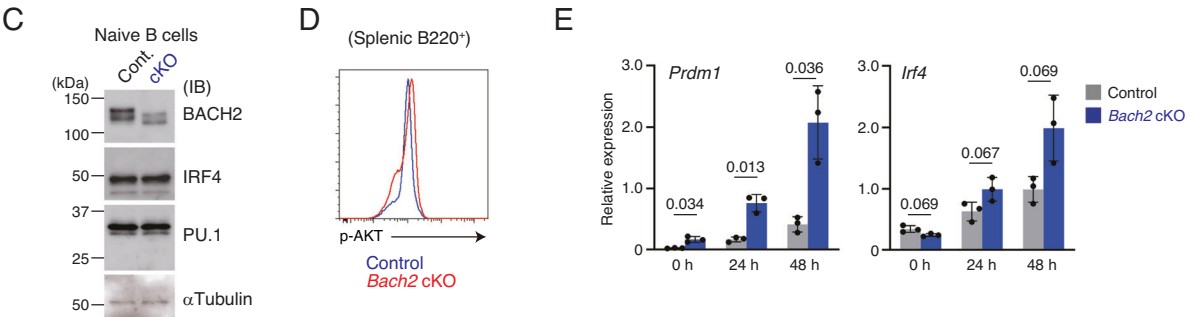

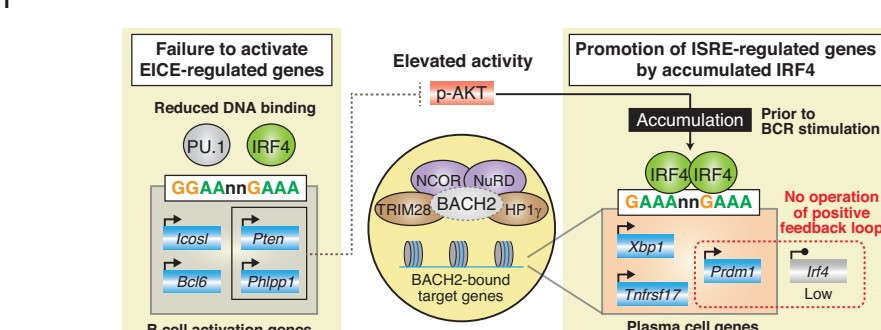

**Two aberrant IRF4 functions bring about
the accelerated plasma cell differentiation in *Bach2⁻ᐟ⁻* B cells**

**Figure 7. Reduced PU.1-IRF4 function and decreased *Irf4* expression occurred in *Bach2⁻/⁻* mice follicular B cells.**

(A) Flow cytometry analysis of splenic B220⁺CD19⁺ cells from *Bach2⁺/⁺* or *Bach2⁻/⁻* mice. FO follicular; MZ marginal zone. The shown plots are representative of three independent experiments. (B) The expression of indicated genes in FO and MZ B cells. (C–E) Analysis of splenic B cells purified from *Bach2ᶠˡ/ᶠˡ* (control) or Mb1-Cre:*Bach2ᶠˡ/ᶠˡ* (*Bach2* cKO). (C) Immunoblot analysis of BACH2, IRF4 and PU.1. αTUBULIN was used as an internal control. kDa kilodalton. (D) Flow cytometry analysis of p-AKT. Data are representative of three independent experiments. (E) Quantification of the *Prdm1* and *Irf4* at the indicated time. Cells were stimulated with LPS and IL-4. (B, E) Data show the average values ± SD (error bars) acquired from three mice for each genotype. *P*-value by *t*-test using R. (F) Schematics of two aberrant functions of IRF4, which bring about accelerated PC differentiation in *Bach2⁻/⁻* B cells. PU.1-IRF4 failed to activate the EICE-regulated genes, including *Icosl, Bcl6, Pten,* and *Phlpp1*. Particularly, the reduced expression of *Pten* and *Phlpp1* presumably contribute to elevated AKT activity, which promotes IRF4 protein accumulation in *Bach2⁻/⁻* B cells. Accumulated IRF4 promotes the ISRE-regulated genes including *Prdm1, Xbp1* and *Tnfrsf17*. These genes are regulated by BACH2-mediated heterochromatin, and the loss of *Bach2* seems to release their gene loci from heterochromatin formation. Importantly, IRF4 was accumulated without increasing *Irf4* mRNA expression, indicating that IRF4-BLIMP-1-positive feedback loop was not operating in these cells. Source data are available online for this figure.

ISRE-regulated gene, was robustly increased in both *Bach2*$^{-/-}$ FO B and MZ B cells, indicating their commitment to PCs. The expression of *Irf4* was significantly decreased in *Bach2*$^{-/-}$ FO B cells, while it was not altered in *Bach2*$^{-/-}$ MZ B cells. These results suggest that *Bach2*$^{-/-}$ FO B cells are driven to PC differentiation by the combination of a decrease in the PU.1-IRF4 function and an increase in the IRF4 homodimer function. Thus, loss of BACH2 facilitates overall PC differentiation of splenic B cells, without activating the IRF4-BLIMP-1-positive feedback loop.

We found that the above aberrant PC differentiation observed in *Bach2*$^{-/-}$ mice was not the case in B-cell-specific *Bach2*-deficient (Mb1-Cre:*Bach2*$^{fl/fl}$, hereafter cKO) mice. In *Bach2* cKO B cells, IRF4 protein was not accumulated (Fig. 7C), and AKT was not activated, accompanied with the unaltered expression of EICE-regulated genes, *Bcl6*, *Icosl*, and *Pten* (Fig. 7D; Appendix Fig. S5A). Upon stimulation with LPS and IL-4, the expression of both *Prdm1* and *Irf4* was induced more highly along the time course in *Bach2* cKO B cells than control B cells (Fig. 7E). The expression of *Aicda* was not induced, and *G1glt* was induced at 24 h but decreased at 48 h in these cells (Appendix Fig. S5B). In flow cytometry analysis, control B cells were enriched with the IRF4 intermediate (IRF4$^{inter}$) population which undergoes CSR (Appendix Fig. S5C). *Bach2* cKO B cells increased the IRF4$^{hi}$ population at early cell division, with lacking CSR. Thus, differing from *Bach2*$^{-/-}$ B cells, PC differentiation was driven by the IRF4-BLIMP-1-positive feedback loop in *Bach2* cKO B cells.

## Discussion

In this study, we have integrated the roles of BACH2 and IRF4 in the promotion of CSR and PC differentiation (Appendix Fig. S6), and how they are altered in the B cell of *Bach2*-deficient mice (Fig. 7F). The duration of BACH2 function determines CSR in activated B cells (Muto et al, 2010; Ochiai et al, 2006), while BACH2 is inactivated by the BCR signaling (Tamahara et al, 2017). We found that PU.1-IRF4 promotes BACH2 function by inducing *Bach2* expression and reducing AKT activity (Fig. 5K–N). In the promotion of PC differentiation, IRF4 is increased at both transcripts and protein levels (Minnich et al, 2016; Sciammas et al, 2006), and AKT enhanced accumulation of IRF4 protein, which led to the expression of the PC genes through their ISREs bound by IRF4 (Ochiai et al, 2013). Combined with our previous reports (Ando et al, 2016; Tamahara et al, 2017), we suggest the presence of "BACH2 on-off system" in B cells undergoing CSR (Appendix Fig. S6). (i) BACH2 represses the CSR- and PC-related gene loci in naive B cell, (ii) the BCR-AKT-mTOR signaling turns off the BACH2 function, (iii) PU.1-IRF4 promotes the BACH2 function in activated B cells, and (iv) increased AKT activity terminates the BACH2 function. Thus, BACH2 and IRF4 cooperatively promote CSR prior to PC differentiation. However, in *Bach2*$^{-/-}$ B cells, PU.1 binding to the PU.1-IRF4 regulatory regions was reduced (Fig. 5I), resulting in the downregulation of EICE-regulated genes including *Pten* and *Phlpp1*, inhibitors of the AKT pathway, as well as *Bcl6* and *Icosl*, GC B-cell genes. Furthermore, IRF4 was accumulated post-transcriptionally (Fig. 3B,C), and induced the expression of PC genes including *Prdm1, Xbp1,* and *Tnfrsf17* by binding to their ISREs (Fig. 4E–G). These observations indicate that *Bach2*$^{-/-}$ B cells were primed for PC differentiation because of the altered balance of EICE- and ISRE-mediated gene regulation by IRF4 (Fig. 7F).

In previous studies, we have shown that BACH2-mediated gene repression involves histone deacetylation by recruiting the NCOR1 and NuRD complexes at its regulatory regions (Ando et al, 2016; Tanaka et al, 2016). Here, we explored additional gene repression machinery by BACH2 in B cells. The BACH2 complexes purified from primary B cells contained the H3K9me3-mediated heterochromatin factors, such as TRIM28, HP1γ and histone H1, as well as the nuclear membrane protein LAMIN B1 (Fig. 1E). BACH2 was co-localized with TRIM28 or HP1γ in the proximity of the nuclear membrane in naive B cells (Fig. 1G,H). In our analysis, the BACH2 complexes did not contain major components of the PRC2 complex which promotes the formation of H3K27me3-medited heterochromatin (Wiles and Selker, 2017). Enhancer of zest 2 (EZH2), the catalytic subunit of the PRC2 complex, is highly induced in GC B cells and represses plasma-cell differentiation (Béguelin et al, 2016; Herviou et al, 2019; Velichutina et al, 2010). While we cannot exclude the involvement of H3K27me3 in BACH2-mediated gene regulation, it was not the case in the cells we examined in this study. Upon B-cell activation, a large number of gene loci lost H3K9me3 modification (Kieffer-Kwon et al, 2017), and a substantial part of these loci was also found to lose BACH2 binding (Fig. 2B). Such genes included CSR-related genes, *Aicda* and *Batf*, as well as PC genes, *Prdm1* and *Xbp1* (Fig. 2B,F,I). The BACH binding motif was predominantly detected within the sequences of the BACH2-binding regions, indicating the involvement of BACH2-mediated heterochromatin formation. In addition to BACH binding motif, BACH2 also binds to the AP-1 motif or AICE (Hipp et al, 2017; Kuwahara et al, 2016). *Batf* is one of the genes negatively regulated by BACH2 binding to the AP-1 motifs at the locus (Kuwahara et al, 2016). *Aicda* is regulated by AICEs (Ochiai et al, 2018), composed of AP-1 and IRF motifs. BACH2 may regulate the heterochromatin formation of the *Batf* and *Aicda* loci via AP-1 motifs or AICEs. Overall, such BACH2 function is turned off by the AKT-mTOR activation via BCR (Tamahara et al, 2017), resulting in release of the BACH2-regulated gene loci from heterochromatin. Then, the expression of *Aicda* is induced by BATF and a lower protein amount of IRF4 (Ochiai et al, 2018), followed by the promotion of CSR. At that time, the expression of *Prdm1* and *Xbp1* is kept at a low level. However, their expression was induced by accumulated IRF4 in *Bach2*$^{-/-}$ B cells (Fig. 4B). On the other hand, the expression of *Batf* and *Aicda* was not induced in *Bach2*-deficient naive B cells (Fig. 3A), suggesting the failure of IRF4-BATF function. Once IRF4 protein level is increased, the balance of gene regulation would be readily shifted towards activation at gene loci modulated by BACH2-binding motifs and ISREs. These observations emphasize the importance of maintaining low IRF4 protein level in B cells undergoing CSR. Taken together, BACH2 works to promote CSR in concert with a lower protein amount of IRF4, and its function is abrogated by IRF4 accumulation.

The positive feedback of IRF4-BLIMP-1 drives PC differentiation (Minnich et al, 2016). However, there is an enigma of how the feedback regulation is initiated; how is IRF4 accumulated to induce *Prdm1* expression? Our study strongly suggests that AKT activation leads to an accumulation of IRF4 protein. The pharmacological inhibition of AKT reduced IRF4 protein level in both activated *Bach2*$^{+/+}$ B cells and *Bach2*$^{-/-}$ B cells (Fig. 6E,F). AKT activity was limited by PU.1 in activated B cells (Fig. 5M), while it was elevated in PCs expressing little *Spi1* mRNA (Fig. 6A) (Carotta et al, 2014). The binding of PU.1 to the EICE-regulated IRF4 target genes was

obviously decreased in *Bach2*$^{-/-}$ B cells (Fig. 5I), presumably resulting in a failure to restrict AKT activity by inducing its negative regulators, including *Pten* and *Phlpp1* (Luo et al, 2019). In consequence, IRF4 protein was accumulated to induce the ISRE-regulated genes including *Prdm1* (Fig. 4B). Thus, the elevated AKT activity facilitated IRF4 protein accumulation, followed by PC differentiation in *Bach2*$^{-/-}$ B cells. Regulation of IRF4 protein by AKT may also be important for memory B cells which differentiate to PCs shortly after BCR stimulation. In memory B cells, IRF4 protein is maintained at a low level (Kometani et al, 2013), and IRF4 instantly commences the PC-GRN when BCR activates the AKT pathway. Considering that AKT activates mTORC1, which positively controls mRNA translation (Ma and Blenis, 2009), the mechanism of IRF4 protein accumulation might involve an increased translation of *Irf4*. Further study is required to explore the mechanism of IRF4 protein accumulation.

In a previous report, we have shown that *Bach2*-deficient mice lacked the germinal center formation upon immunization (Muto et al, 2004). BCL6, a critical regulator of GC formation (Basso and Dalla-Favera, 2010), is expressed in naive B cells (Muto et al, 2010) and represses *Prdm1* with BACH2 cooperatively (Ochiai et al, 2008). The expression is highly induced upon BCR stimulation by PU.1-IRF4 (Ochiai et al, 2013). Considering that the transcriptional function of PU.1 is activated by AKT (Rieske and Pongubala, 2001), PU.1-IRF4 regulate the expression of *Bcl6* in both naive B cells and activated B cells. The reduced DNA binding of PU.1 presumably resulted in the downregulation of *Bcl6*, followed by the failure of GC formation in *Bach2*-deficient mice. Including *Bcl6* and *Pten*, the expression of EICE-regulated genes was reduced in FO B cells but not in MZ B cells in *Bach2*$^{-/-}$ B cells (Fig. 7B), indicating the reduced PU.1 functions in *Bach2*$^{-/-}$ FO B cells. It will be necessary to understand the alteration of the transcriptome as well as the PU.1 activity in *Bach2*$^{-/-}$ FO B cells and MZ B cells.

Importantly, antibodies produced in *Bach2*$^{-/-}$ mice sera are autoreactive (Jang et al, 2019; Roychoudhuri et al, 2013). BACH2 regulates proper differentiation of not only B cells but also T cells (Ebina-Shibuya et al, 2017; Roychoudhuri et al, 2013), NK cells (Li et al, 2022) and dendritic cells (Kurotaki et al, 2018). Compared with *Bach2*$^{-/-}$ mice, *Bach2* cKO mice showed less abnormalities in B cells (Fig. 7C–E; Appendix Fig. S5A–C) and no production of autoreactive antibodies (Jang et al, 2019). Remarkably, the dysfunction of BACH2 in CD4$^+$ T$_{reg}$ differentiation leads to autoreactivity in *Bach2*$^{-/-}$ mice, accompanied with the production of inflammatory cytokines (Kuwahara et al, 2016; Roychoudhuri et al, 2016; Roychoudhuri et al, 2013; Trujillo-Ochoa et al, 2023). These observations strongly suggest that the involvement of non-B cells in the abnormalities in *Bach2*$^{-/-}$ B cells. Inflammatory cytokines activate various cell signaling including the AKT pathway via cytokine receptors, and B-cell abnormality in *Bach2*$^{-/-}$ mice is presumably brought about by the inflammatory milieu caused by the integrated effect of immune cell dysfunction. BACH2 dysregulation may also confer disease vulnerability, including autoimmune diseases (Afzali et al, 2017; Nakano et al, 2022; Zhou et al, 2023), by inducing IRF4 protein accumulation as well as the chromatin dissociation of PU.1 in B cells. Exploring how cell signaling modulates the function of these key TFs will provide a clue to prevent the production of autoreactive antibodies from B cells under aberrant cell environment.

Collectively, our study describes the reciprocal regulation between BACH2 and IRF4 in the course of PC differentiation and demonstrates that *Bach2*-deficiency accelerates PC differentiation by altering the regulations of IRF4 and its partner TF PU.1 in B cells. BACH2 is a critical regulator for the acquisition of antigen-specific antibodies, and such BACH2 function is regulated by cellular molecules including mTORC1 and heme (Ando et al, 2016; Tamahara et al, 2017; Watanabe-Matsui et al, 2011). The mechanism for the reactivation and maintenance of BACH2 could be exploited in both B cells and other immune cells to increase or to mitigate immune response.

## Methods

### Mice

*Bach2*-deficient mice were described previously (Muto et al, 2004). B1-8$^{hi}$ mice (Shih et al, 2002) were from M. Nussenzweig (Rockefeller University), Mb1-Cre (Hobeika et al, 2006) mice and *Bach2*$^{flox/flox}$ mice (Kometani et al, 2013) were from and T. Kurosaki (Osaka University, RIKEN). B1-8$^{hi}$:*Bach2*$^{-/-}$ mice and Mb1-Cre:*Bach2*$^{flox/flox}$ mice were generated, and they were born at the expected Mendelian ratio with no obvious abnormality. All mice were maintained in pathogen-free conditions in accordance with guidelines approved by the Institution for Animal Experimentation Committee at Tohoku University Graduate School of Medicine (2019MdA-218, 2021MdA-098). Experiments were performed using sex- and age-matched mice between 8 and 12 weeks of age.

### In vitro plasma-cell differentiation

Splenic B cells were isolated using the B-cell isolation kit and LS columns (Miltenyi Biotec), and cultured in RPMI-1640 (Sigma) media supplemented with 10% FBS, 10 mM HEPES, 1 mM Na-Pyruvate, 0.1 mM non-essential amino acids, 100 U/mL penicillin, 100 μg/mL streptomycin, 50 μM β-mercaptoethanol. Splenic B cells purified from B1-8$^{hi}$ background mice were stimulated with recombinant mouse IL-2 (100 U/mL; R&D Systems), recombinant mouse IL-4 (5 ng/mL; BD Biosciences), recombinant mouse IL-5 (1.5 ng/mL; R&D Systems), recombinant mouse CD40L (0.2 ng/mL) (R&D Systems), and NP (4-Hydroxy-3-nitrophenylacetic)$_{40}$-ficoll (0.01 ng/mL) (Biosearch Technologies Inc.). Splenic B cells purified from *Bach2*$^{flox/flox}$ mice were stimulated with LPS (20 μg/mL; Sigma) and recombinant mouse IL-4 (10 ng/mL; BD Biosciences). AKT inhibitor AZD5363 (Capivasertib; MedChem-Express) was dissolved in DMSO, and added in culture medium.

### Immunoblot analysis and Immunohistochemistry

Whole-protein extracts were prepared using RIPA extraction buffer (50 mM Tris-HCl (pH 8.0), 0.1% SDS, 150 mM NaCl, 0.02% NaN$_3$, 1% NP-40, 0.5% sodium deoxycholate) supplemented with protease inhibitor and phosphatase inhibitor cocktails (Roche) as described (Ochiai et al, 2020). Lysates were separated by SDS-PAGE and transferred to Immobilon-P membranes (Millipore). Antibodies used in this study were as follows: anti-BACH2 (N1; homemade), anti-mTOR-p (S2448) (#2971; Cell Signaling), anti-mTOR (#2972; Cell Signaling), anti-AKT-p (S473) (#9271; Cell Signaling),

anti-AKT (#9272; Cell Signaling), anti-TRIM28/TIF1β/KAP1 (sc-33186; Santa Cruz), anti-HP1γ (#2619; Cell Signaling), anti-IRF4 (sc-6059; Santa Cruz), anti-PU.1 (sc-352; Santa Cruz), anti-αTUBULIN (sc-5286; Santa Cruz) and anti-β-ACTIN (GTX109639; GeneTex). The secondary antibodies used were horseradish peroxidase (HRP)-conjugated anti-rabbit IgG, HRP-conjugated anti-mouse IgG, and HRP-conjugated anti-goat IgG (GE Healthcare). Immunohistochemistry was performed using anti-BACH2 (N1; homemade, sc-14702; Santa Cruz), anti-LAMIN B1 (sc-6217; Santa Cruz), anti-TRIM28/TIF1β/KAP1 (sc-33186; Santa Cruz) or anti-HP1γ (#2619; Cell Signaling) antibodies as described (Tamahara et al, 2017). Data were obtained using LSM780 confocal microscope system (ZEISS).

## Complex purification and LC-MS/MS analysis

B1-8$^{hi}$ splenic B cells were stimulated for 12 h, and BACH2 complexes were purified using ReCLIP (Ochiai et al, 2021; Smith et al, 2011). Cells were crosslinked with reversible crosslinkers, 0.5 mM DTME (Thermo Scientific) and 0.5 mM DSP (Thermo Scientific), at 25 °C for 30 min. After removal of the crosslink buffer, cells were quenched with 20 mM Tris (pH 7.5) 5 mM Cystein at 25 °C for 5 min, followed by washing with ice-cold PBS twice. Cells were suspended in RIPA buffer supplemented with proteinase inhibitor cocktail and PhosSTOP (Roche), and lysed on ice for 10 min. The cell lysate was sonicated using Bioruptor (Cosmo Bio), and centrifuged. The supernatant was reacted with protein A and G beads (Invitrogen) at 4 °C for 1 h for removal of non-specific reaction. After beads trap, the lysate was reacted with conjugated anti-BACH2 (N1; homemade) antibodies or control IgG (sc-2027; Santa Cruz) for immunoprecipitation at 4 °C overnight. Beads were washed with chilled RIPA buffer three times, and eluted with elution buffer (50 mM Tris (pH 8.0), 0.2 M NaCl, 2% SDS and 125 mM DTT) at 37 °C for 20 min, then 70 °C for 10 min. The tryptic peptides were analyzed to determine each protein using LTQ OrbiTrap Velos (Thermo Fisher Scientific) and the MASCOT search engine (Matrix Science) as previously described (Hipp et al, 2017).

## Flow cytometry

Bone marrow or spleen cells were collected and suspended in FACS buffer (1% FBS in PBS) and subjected to the surface or intracellular staining as described previously (Ochiai et al, 2013; Ochiai et al, 2020). Cells were stained with anti-B220 (RA3-6B2; eBioscience), anti-CD19 (1D3: BD Biosciences), anti-CD43 (S7; BD Biosciences), anti-IgM (II/41; eBioscience), anti-IgD (11–26 c.a; BD Biosciences), anti-IgG1 (A85-1; BD Biosciences), anti-CD138 (281-2; BD Biosciences), CD21 (7G6; BD Biosciences), CD23 (B3B4; BD Biosciences) and anti-Streptavidin-PerCP (BD Biosciences) antibodies. For intracellular staining, cells were fixed with 1% paraformaldehyde for 10 min and washed twice with FACS buffer. Cells were stained with anti-IRF4 (sc-6059; Santa Cruz) or p-AKT (S473) (R&D systems) antibody in 0.3% Saponin/PBS. For IRF4 staining, cells were incubated with goat anti-IRF4 antibodies in the presence of 5% donkey serum, followed by a Cy5-coupled donkey anti-goat (712-606-153; Jackson Immunoresearch). For cell division trace assay, the isolated B cells were loaded with 5 μM cell trace violet (C34557; Thermo Fisher Scientific, Molecular Probes) according to the manufacturer's instructions prior to culturing. Data were collected with the FACS Aria II or FACS verse (BD Biosciences) and analyzed with FlowJo software (TreeStar).

## Retroviral vectors and transduction in activated splenic B cells

Retroviral vector murine PU.1 (MSCV-EGFP-PU.1WT) has been described (Laslo et al, 2006). MSCV-dsRedT4-PU.1WT was constructed by cloning PU.1WT from MSCV-EGFP-PU.1WT into the MSCV-dsRedT4. Virus supernatants were prepared and infected to activated B cells after 18–20 h of in vitro stimulation as previously described (Ochiai et al, 2012). Transduced cells were sorted based on GFP or dsRedT4 expression after 48 h, followed by flow cytometry analysis or cell collection for RNA extraction.

## RNA isolation and RT-qPCR

Total RNA was prepared using an RNeasy Plus Mini Kit (Quiagen), and cDNA was made with Superscript III reverse transcriptase (Invitrogen). The FastStart SYBR Green Master SYBR Green I (Roche) for cDNA was used for quantitative PCR by LightCycler96 (Roche). The relative expression was normalized by β2-microglobulin (β2 m). Primers used in this study are shown in Appendix Table S2.

## RNA-seq

Splenic B cells were purified from B1-8$^{hi}$:*Bach2*$^{+/+}$ or B1-8$^{hi}$:*Bach2*$^{-/-}$, and total RNA was prepared from triplicate cell samples using an RNeasy Plus Mini Kit (Quiagen). Each library was prepared using a TruSeq RNA sample preparation kit v2 (Illumina), and the libraries were clonally amplified on the flow cell and sequenced on an Illumina HiSeq2500 with a 51-mer paired-end sequence. Image analysis and base calling were performed using Real-Time Analysis (RTA) 1.13. The RNA-seq reads were aligned to a mouse reference genome (mm10/GRCm38), using TopHat (version 2.2.3) with all parameters set to default (Trapnell et al, 2012). The expression levels were calculated using the Cuffdiff module of the Cufflinks program (version 2.2.3) (Trapnell et al, 2012).

## Chromatin immunoprecipitation (ChIP) and ChIP-seq

Chromatin was isolated from activated B cells and sonicated to obtain DNA fragments ranging in size from 100 to 500 base pairs using the Bioruptor (BMBio). Chromatin fragments were immunoprecipitated with Dynabeads Protein A and G beads (Invitrogen/Thermo Fisher Scientific) conjugated with anti-BACH2 (Homemade) or anti-H3K27ac (ab4729; Abcam) antibodies at 4 °C overnight. Beads were washed with wash buffer 1 (0.1% SDS, 1% Triton-X 100, 2 mM EDTA, 20 mM Tris-HCl pH 8.1, 150 mM NaCl), 2 (0.1% SDS, 1% Triton-X 100, 2 mM EDTA, 20 mM Tris-HCl pH 8.1, 500 mM NaCl) and 3 (1% NP-40, 250 mM LiCl, 1 mM EDTA, 10 mM Tris pH 8.1), then Tris-EDTA (pH 8.0) twice. DNA was eluted using elution buffer (1% SDS, 100 mM NaHCO$_3$), reacted for 15 min at 25 °C, twice. Two elutions were combined, and NaCl (final 0.2 M) and RNase A (final 0.4 mg/mL) were added. Reverse crosslinks were performed at 65 °C overnight. After reacted with Proteinase K buffer (final 4 μg/mL Proteinase K, 20 mM EDTA, 40 mM Tris pH 6.5) at 45 °C for 1 h, DNA was purified using a DNA purification Kit (Qiagen). For

ChIP-seq alignment, ChIP-seq and input DNA reads were mapped on a mouse reference genome sequence (mm10/GRCm38) using Bowtie software (bowtie2-2.3.2). ChIP-seq peak identification was performed using HOMER package (version v4.11) (Heinz et al, 2010). Mapped SAM format files were converted to tag directories using the makeTagDirectory module, and ChIP-seq peaks were identified by using the findPeaks module with "-style factor" and "-style histone" options for transcription factor and histone modifications, respectively. DNA-binding motif of transcription factors were identified using the findMotifsGenome .pl module with the default parameter. For the visualization of ChIP-seq tags, the bedGraph files were generated using the makeUSCSfile module, and uploaded on the UCSC genome browser (https://genome.ucsc.edu/). The distribution of histone modification was analyzed using Galaxy plotHeatmap (https://usegalaxy.org/). For ChIP-qPCR, primers used in this study are shown in Appendix Table S2.

## Quantification and statistical analysis

Bar graphs for quantitative RT-PCR and ChIP-PCR were drawn using GraphPad Prism9. Data are represented as average, and error bars indicate standard deviation. Statistical significance was determined with Welch two sample $t$-test using the open-source statistical programing environment R (version 3.4.0).

## Data availability

We utilized our previous sequencing data for input reads, IRF4 ChIP-seq day 3 (as 72 h), and H3K27ac ChIP-seq (as $Bach2^{+/+}$) prepared from B1-8$^{hi}$ mice splenic B cells (GEO: GSE145951). Additional new RNA-seq and ChIP-seq data have been uploaded to the NCBI Gene Expression Omnibus (PRJNA998173; https://dataview.ncbi.nlm.nih.gov/object/PRJNA998173?reviewer=nvjl7ink06nkdn57td3d181bsi).

## Peer review information

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

## Acknowledgements

The authors thank A Muto and the members of the Department of Biochemistry, Tohoku University Graduate School of Medicine, for the discussions; M Kikuchi (Tohoku University) for operating the Illumina HiSeq2500. This work is supported by Japan Society for the Promotion of Science grants-in-aid (16K19026, 20K07351, and 23H02671 to KO, 20K07321 to HS, 18H04021 and 22H00443 to KI), the Agency for Medical Research and Development (AMED-CREST 16gm050001 to KI), Life Science Foundation of Japan (KO) and Research Grant in the Natural Sciences from the Mitsubishi Foundation (KI). Part of this study was supported by the Biomedical Research Core of the Tohoku University School of Medicine.

## Author contributions

**Kyoko Ochiai**: Conceptualization; Data curation; Formal analysis; Supervision; Funding acquisition; Validation; Investigation; Visualization; Methodology; Writing—original draft; Project administration; Writing—review and editing. **Hiroki Shima**: Funding acquisition; Investigation. **Toru Tamahara**: Investigation. **Nao Sugie**: Investigation. **Ryo Funayama**: Resources. **Keiko Nakayama**: Resources. **Tomohiro Kurosaki**: Resources. **Kazuhiko Igarashi**: Data curation; Supervision; Funding acquisition; Writing—review and editing.

## Disclosure and competing interests statement

The authors declare no competing interests.

