## [Peer Review File · The EMBO Journal]

Accelerated plasma-cell differentiation in Bach2-deficient mouse B cells is caused by altered IRF4 functions

Kyoko Ochiai, Hiroki Shima, Toru Tamahara, Nao Sugie, Ryo Funayama, Keiko Nakayama, Tomohiro Kurosaki, and Kazuhiko Igarashi

Corresponding authors: *Kyoko Ochiai (kochiai@med.tohoku.ac.jp)* , *Kazuhiko Igarashi (igarashi@med.tohoku.ac.jp)*

Review Timeline:

Submission Date:	13th Sep 23
Editorial Decision:	20th Nov 23
Revision Received:	8th Jan 24
Editorial Decision:	13th Feb 24
Revision Received:	19th Feb 24
Accepted:	24th Feb 24

Editor: Kelly Anderson

Transaction Report:

Dear Dr. Ochiai,

Thank you for submitting your manuscript for consideration by the EMBO Journal. It has now been seen by three referees whose comments are shown below.

Given the referees' positive recommendations, I would like to invite you to submit a revised version of the manuscript, addressing the comments of all three reviewers. It is EMBO Journal policy to allow only a single round of revision, and acceptance of your manuscript will therefore depend on the completeness of your responses in this revised version. It would be good to discuss your plan to address the referee concerns and I am available to do so by zoom or email in the coming weeks.

Thank you for the opportunity to consider your work for publication. I look forward to your revision.

Yours sincerely,

Kelly M Anderson, PhD
Editor, The EMBO Journal
k.anderson@embojournal.org

We realize that it is difficult to revise to a specific deadline. In the interest of protecting the conceptual advance provided by the work, we recommend a revision within 3 months (18th Feb 2024). Please discuss the revision progress ahead of this time with the editor if you require more time to complete the revisions.

Referee #1:

In this manuscript, the authors go beyond their previous analyses of the mechanisms supporting BACH2-mediated gene repression in B cells. This new study documents that BACH2 associates with TRIM28, HP1 γ , histone H1, LAMIN B1, likely mediating heterochromatin formation proximal to the nuclear membrane in naïve cells. The pattern of BACH binding is then affected by B-cell activation and lost for *Aicda*, *Batf*, *Prdm1* and *Xbp1*. The study also shows that BACH2 function is switched off by the AKT-mTOR cascade for expression of *Aicda*, together with a low level of *Prdm1* and *Xbp1* maintained as long as B cells are undergoing CSR. Later, accumulated IRF4 abrogates the function of BACH2, while PC differentiation can occur. Altogether, combining functional assays with immunoprecipitation and LC-MS/MS, the authors provide new and important insights into the cross-regulation of BACH2 and IRF4 and into the remodeling of major transcription factor networks in the course of B-cell activation and PC differentiation.

Referee #2:

In this manuscript, Ochiai et al examine the roles of Bach2 and IRF4 in driving class switching vs plasma cell differentiation. The authors demonstrate that Bach2 influenced the chromatin architecture of IRF-regulated genes (*Aicda*) and *Prdm1*. Interestingly, they find that in Bach2-deficient B cells, genes modulating plasma cell differentiation are activated by the accumulated IRF4, without an increase in *Irf4* mRNA, possibly via reduced activity of the PU.1-IRF4 complex, which in turn induces the expression of Bach2 and *Pten*, a negative regulator of AKT signaling. Importantly, AKT activity was elevated in Bach2-deficient B cells, resulting in IRF4 protein accumulation. The authors conclude that BACH2 and IRF4 mutually modulate the function of each other and that Bach2 inhibits plasma cell differentiation by both repressing plasma cell genes and restricting accumulation of IRF4. The mechanisms that govern class switching vs plasma cell differentiation are still poorly understood and the roles of Bach2 and IRF4 in this process has not been fully elucidated. The study is thus of much relevance to the B cell biology. However, the manuscript lacks several controls as outlined below (and extremely difficult to follow).

The authors should address the following issues (and rewrite the manuscript to make it more reader-friendly) to raise the impact of the work.

Fig 1. The authors should use Bach2 deficient cells as a negative control for background proteins, not non-specific antibodies. The authors are cross-linking the cells and without the right controls, the chances of picking up non-specific proteins is quite high.

Fig. 2. The authors have essentially analyzed the data generated by others. Are the activation conditions similar between this previous work and that done by the authors here? And more importantly, is there a control for Bach2 binding, as in a ChIP with the Bach2 antibody from Bach2-deficient cells? What are the relative levels of Bach2 in pro-pre-B cells vs activated B cells? It is extremely difficult to interpret the results shown in fig 2.

Fig. 3. Why do the authors pick up *Aicda* in naïve B cells? It is not clear how the authors are purifying the B cells from the mouse spleen, just looking at the data it is possible that the purification is not clean and there are activated B cells in this population. Is there any specific reason why they have used the B1-8 mouse for this particular analysis? It is also quite surprising that the authors can detect *Bcl6* expression in the naïve B cells. The authors should confirm this expression by repurifying bona fide naïve B cells (maybe via CD43-ve purification). Again it is unclear how they have purified the naïve B cells here.

Fig. 7. Are the experiments done in B cells derived from immunized mice? If not, it is surprising to find *Bcl6* expression (as mentioned above). What is the frequency of GCs in these spleen? Additionally, *mb1-cre* will delete Bach2 in early B cells. Is there a developmental defect? What is the rationale for using B1-8 mice?

Referee #3:

In this manuscript, the authors investigated how the roles of BACH2 and IRF4 were integrated into the regulation of CSR and PC differentiation. The authors reported that BACH2 organized heterochromatin formation of target genes including IRF4 target genes and *PRDM1*. They reported that PU.1-IRF4 promotes BACH2 function by inducing BACH2 expression and reducing AKT activity. They demonstrated that BACH2 and IRF4 mutually modulate their biological functions and that BACH2 inhibits PC

differentiation repressing PC genes and the restriction IRF4 protein accumulation.

The paper is well written with a large data set of experiments. Data analysis has been thoroughly performed and support the conclusion. Supplemental data ideally complement main figures.

I can come with some hopefully constructive comments:

1/ The heme was shown to directly bind and inhibit BACH2 function (Watanabe-Matsui M et al. Blood 2011; Jang KJ et al. Nat Comm 2015). Is there a role of heme in the presented results?

2/ Inhibition of Polycomb PRC2 complex was shown to accelerate PC differentiation (Herviou L et al. Leukemia 2019). Did the authors investigated the involvement of H3K27me3 in BACH2-mediated target gene repression?

3/ Early UPR activation mediated by mTORC1 was reported in activated B cells (Gaudette BT et al. Nat Comm 2020) driving PC priming before XBP1 gene expression. The authors could investigate how it could be affected in BACH2^{-/-} cells.

4/ Regarding the figure 7, it would be interesting to have a complete transcriptome analysis of Bach2^{-/-} FO and MZ B cells.

5/ The analysis of PC differentiation comparing BACH2^{-/-} and BACH2^{+/+} B cells at the single cell level could be important to understand how the maturation trajectories are affected.

5/ The authors could discuss more the results presented in Figures 5 C, D and E demonstrating that results obtained with the BACH2^{-/-} B cells are not identified in B cell specific Bach2 depletion model.

Minor comments:

P16: typo error.

Response to reviewer comments

EMBOJ-2023-115594 K. Ochiai et al.

We really appreciate the opportunity to re-submit our revised manuscript to The EMBO Journal. The comments from the three reviewers became guide to reorganize our manuscript. The comments of each reviewer are addressed in revised manuscript, as well as on the document named "Response to reviewer comments". Major changes we made are summarized as following.

- 1) To improve the description how we extracted genomic regions which lost BACH2 binding upon B cell activation,

We have added Figure S1, the *de novo* motif analysis of BACH2 ChIP-seq in *Ebf1*-deficient pre-pro-B cells or activated B1-8^{hi} splenic B cells.

We have modified Table S2 to include all genes bound by BACH2 in pre-pro-B cells and/or activated B cells.

The manuscript has been modified including these additional data which support the interpretation of our strategy to extract genes lost BACH2 binding upon B cell activation.

- 2) We have added RT-PCR data of selected EICE-regulated genes, *Bcl6*, *Icosl* and *Pten*, examined in control and *Bach2* cKO B cells as Revised Figure S5A. The expression of these genes was not altered in *Bach2* cKO B cells, supporting no change of AKT activity and PU.1-IRF4 function in these cells. The description has been added in the manuscript.
- 3) To reflect comments from reviewer #2 and reviewer #3, we have added one paragraph to the discussion (Page 17, line 511). This paragraph discusses about the regulation of *Bcl6* expression, the phenotype of *Bach2*^{-/-} mice, and the direction of future research.
- 4) To reply reviewer #2, we have provided additional supplemental data as a separate PDF file named "DataReplyToReviewer2".
- 5) We have modified the manuscript to reflect reviewer's comments.

Referee #1:

In this manuscript, the authors go beyond their previous analyses of the mechanisms supporting BACH2-mediated gene repression in B cells. This new study documents that BACH2 associates with TRIM28, HP1 γ , histone H1, LAMIN B1, likely mediating heterochromatin formation proximal to the nuclear membrane in naïve cells. The pattern of BACH binding is then affected by B-cell activation and lost for Aicda, Batf, Prdm1 and Xbp1. The study also shows that BACH2 function is switched off by the AKT-mTOR cascade for expression of Aicda, together with a low level of Prdm1 and Xbp1 maintained as long as B cells are undergoing CSR. Later, accumulated IRF4 abrogates the function of BACH2, while PC differentiation can occur.

Altogether, combining functional assays with immunoprecipitation and LC-MS/MS, the authors provide new and important insights into the cross-regulation of BACH2 and IRF4 and into the remodeling of major transcription factor networks in the course of B-cell activation and PC differentiation.

REPLY: We would like to thank to this reviewer for evaluating the manuscript.

Referee #2:

In this manuscript, Ochiai et al examine the roles of Bach2 and IRF4 in driving class switching vs plasma cell differentiation. The authors demonstrate that Bach2 influenced the chromatin architecture of IRF-regulated genes (*Aicda*) and *Prdm1*. Interestingly, they find that in Bach2-deficient B cells, genes modulating plasma cell differentiation are activated by the accumulated IRF4, without an increase in *Irf4* mRNA, possibly via reduced activity of the PU.1-IRF4 complex, which in turn induces the expression of Bach2 and *Pten*, a negative regulator of AKT signaling. Importantly, AKT activity was elevated in Bach2-deficient B cells, resulting in IRF4 protein accumulation. The authors conclude that BACH2 and IRF4 mutually modulate the function of each other and that Bach2 inhibits plasma cell differentiation by both repressing plasma cell genes and restricting accumulation of IRF4.

The mechanisms that govern class switching vs plasma cell differentiation are still poorly understood and the roles of Bach2 and IRF4 in this process has not been fully elucidated. The study is thus of much relevance to the B cell biology. However, the manuscript lacks several controls as outlined below (and extremely difficult to follow). The authors should address the following issues (and rewrite the manuscript to make it more reader-friendly) to raise the impact of the work.

REPLY: We would like to thank this reviewer for pointing out important issues. We have addressed each point as below. We have also addressed the structure of the manuscript for easier understanding. For example,

- REPLY to Q1: We have added the description how the components of BACH2 complexes were determined (Supplemental data for Q1).
- REPLY to Q2: We have modified the manuscript to explain detail about how we extracted genes which lost BACH2 binding upon B cell activation.
- REPLY to Q3: We have confirmed that B cells utilized in this study were CD43-negative/B220-positive naïve B cells (Supplemental data for Q3).
- REPLY to Q4: We have provided the data of B cell development and the observations of inflammation comparing *Bach2*-deficient mice and *Bach2* cKO mice (Supplemental data for Q4).

1) Fig 1. The authors should use Bach2 deficient cells as a negative control for background proteins, not non-specific antibodies. The authors are cross-linking the cells and without the right controls, the chances of picking up non-specific proteins is quite high.

REPLY:

[1] Why we used the sample immunoprecipitated using control IgG as a negative control, and how we determined the components of BACH2 complex.

It is ideal to compare the protein complex purified using wild-type cells and cells lacking the protein. However, *Bach2*-deficient B cells were primed for PC differentiation even without antigen stimulation (Figure 4B), and had the different cell feature, such as increased cell size dimension (Supplemental data for Q1), compared with wild-type B cells. Therefore, we used cell extracts prepared from wild-type cells, and processed in one tube. After pre-clearing using protein A/G beads, the samples were divided into two tubes to immunoprecipitate using anti-BACH2 antibodies or control IgG. Each sample was analyzed using LC-MS/MS, and the components of BACH2 complex were determined as specific detection with anti-BACH2 antibodies, or more than two-fold protein score with anti-BACH2 antibodies than control IgG. Furthermore, independently purified three BACH2 complexes were compared, and the components detected in more than two BACH2 complexes were used in this study (Figure 1C).

We have added the description in the manuscript as below.

(Page 6, line 160)

“To explore this possibility, endogenous BACH2 was purified using anti-BACH2 antibodies from B1-8^{hi} mice B cells at 12 hours after BCR stimulation, and interacting proteins were identified using mass spectrometry. **Then, the components of BACH2 complex were determined as specific detection with anti-BACH2 antibodies, or more than two-fold protein score with anti-BACH2 antibodies than control IgG. Furthermore, independently purified three BACH2 complexes were compared, and 80 factors were commonly detected in three BACH2 complexes (Figure 1C). These factors were enriched with GO terms related to “chromatin remodeling”, “DNA repair” and “transcripts” (Figure 1D, Table S1).”**

[2] Explanation for crosslinking the cells for complex purification.

For complex purification, we treated cells with crosslinkers 0.5 mM DTME and 0.5 mM DSP at 25°C for 30 min. The crosslink using these reagents stabilizes the protein-protein interaction in an irreversible manner, allowing us to detect labile complexes with low background (Smith A.L. et al. PLOS One 2011).

The reference was missing and added to Materials and Methods as below.

(Page 26, line 766)

“B1-8^{hi} splenic B cells were stimulated for 12 hours, and BACH2 complexes were purified using ReCLIP^{36, 68}. Cells were crosslinked with reversible crosslinkers, 0.5 mM DTME (Thermo Scientific) and 0.5 mM DSP (Thermo Scientific), at 25°C for 30 min.”

68. Smith, A.L., Friedman, D.B., Yu, H., Carnahan, R.H. & Reynolds, A.B. ReCLIP (reversible cross-link immuno-precipitation): an efficient method for interrogation of labile protein complexes. *PloS one* 6, e16206 (2011).

2) Fig. 2. The authors have essentially analyzed the data generated by others. Are the activation conditions similar between this previous work and that done by the authors here? And more importantly, is there a control for Bach2 binding, as in a ChIP with the Bach2 antibody from Bach2-deficient cells? What are the relative levels of Bach2 in pro-pre-B cells vs activated B cells? It is extremely difficult to interpret the results shown in fig 2.

REPLY: In Figure 2, we extracted genes related to BACH2-mediated heterochromatin. For this purpose, we used two sets of ChIP-seq: H3K9me3 ChIP-seq and BACH2 ChIP-seq.

[1] Usage of H3K9me3 ChIP-seq data sets from Vian L. et al. Cell 2018.

First, we extracted genomic regions which lost H3K9me3 modification, a marker of heterochromatin, upon B cell activation. H3K9me3 ChIP-seq data were referenced from Vian L. et al. Cell 2018, in which naïve B cells were stimulated using LPS and IL-4 with anti-CD180. In a previous report, we have reported that BACH2 is required for CSR in activated B cells stimulated with LPS and IL-4 (Muto A. et al. EMBO J 2010). Therefore, their H3K9me3 ChIP-seq data are supposed to contain BACH2 regulatory

regions losing the modification upon B cell activation.

We have modified the manuscript as below.

(Page 7, line 182)

“It was reported that the number of chromatin regions with enriched H3K9me3 became less in activated B cells than naïve B cells²⁰. They utilized LPS, IL-4 and anti-CD180 antibodies for B cell activation. **In a previous report, we have reported that BACH2 is required for CSR in activated B cells stimulated with LPS and IL-4 (ref.¹⁶). Therefore, their H3K9me3 ChIP-seq data are supposed to contain BACH2 regulatory regions losing the modification upon B cell activation.**”

[2] Interpretation of how we extracted genomic regions which lost BACH2 binding upon B cell activation.

Next, we extracted genomic regions which lost BACH2 binding upon B cell activation. To extract them, we compared BACH2 ChIP-seq data performed in *Ebfl1*-deficient pre-pro-B cells and activated B cells. *Ebfl1*-deficient pre-pro-B cells were established from bone marrow of *Ebfl1*-deficient mice (Pongubala J. Nature Immunology 2008), and we have explored BACH2 binding target genes in these cells (Itoh-Nakadai A. et al. Cell Reports 2017). BACH2 ChIP-seq in activated B cells was newly obtained in this study. Although we have not performed BACH2 ChIP-seq using *Bach2*-deficient B cells, sequences of BACH2-bound regions were enriched for BACH binding motif with the significant P-value (Revised Figure S1). The overall BACH2 binding signal was higher in *Ebfl1*-deficient pre-pro-B cells than activated B cells. BACH motif was detected as the top motif in both BACH2 ChIP-seq data with the similar, low percentage of background. Importantly, more than half, 55.50%, of targets contained the BACH motif in activated B cells, while about one quarter, 26.88%, of targets contained it in pre-pro-B cells (Revised Figure S1). These results indicated that BACH2 ChIP-seq in activated B cells effectively exhibits BACH2 target genes, even though much fewer peaks than pre-pro-B cells were identified (Figure 2B, Revised Table S2).

Importantly, many of the genes related to CSR or plasma cell showed BACH2 binding in pre-pro-B cells but not in activated B cells. On the other hand, genes related to hematopoietic progenitors, non-B lineage immune cells or early B cells showed BACH2 binding in both pre-pro-B cells and activated B cells, and these genes included *Kit*, *Ly96*, *Cish* and *Cxcr4* (Revised Table S2: the lists of genes were marked with blue). These

observations indicated that BACH2 represses the expression of non-B cell genes in both early B cells and activated B cells, and only a selected set of genes were de-repressed upon B cell activation. Therefore, for further analysis, we applied genes bound by BACH2 in pre-pro-B cells but not in activated B cells as a loss of BACH2 binding in activated B cells.

We have added a one paragraph to elaborate the above description in the manuscript. The *de novo* motif analysis of each BACH2 ChIP-seq has shown as revised Figure S1. (Page 7, line 199)

“To identify BACH2 target genes from those genes, we utilized two data sets of BACH2 ChIP-seq. One was our previous data obtained in *Ebf1*-deficient pre-pro-B cells³⁸. A total of 12,128 peaks, corresponding to 8,790 genes, were extracted **with enriched in BACH motif** (Figure 2B, Table S2, **Figure S1**). The other data set was newly obtained from activated B1-8^{hi} B cells. Compared with pre-pro-B cells, much fewer 664 peaks, corresponding to 627 genes, were identified (Figure 2B, Table S2). **Importantly, sequences obtained from BACH2 ChIP-seq in activated B cells were also enriched in BACH motif with the significant P-value, and more than half, 55.50%, of targets contained the BACH motif (Figure S1). These results suggested that BACH2 ChIP-seq in activated B cells effectively exhibits BACH2 binding genomic regions. Importantly, many of the genes related to CSR or plasma cell showed BACH2 binding in pre-pro-B cells but not in activated B cells. On the other hand, genes related to hematopoietic progenitors, non-B lineage immune cells or early B cells showed BACH2 binding in both pre-pro-B cells and activated B cells, and these genes included *Kit*, *Ly96*, *Cish* and *Cxcr4* (Table S2). These observations indicated that BACH2 represses the expression of non-B cell genes in both early B cells and activated B cells, and only a selected set of genes were de-repressed upon B cell activation. Therefore, for further analysis, we applied genes bound by BACH2 in pre-pro-B cells but not in activated B cells as a loss of BACH2 binding in activated B cells.”**

[3] Interpretation of extracted 6,957 genes as lost both H3K9me3 modification and BACH2 binding in activated B cells.

Based on [1] and [2], we have extracted 6,957 genes which lost both H3K9me3 modification and BACH2 binding in activate B cells (Figure 2B). These genes included known BACH2 target genes related to plasma cell differentiation, such as *Prdm1* and

Ccnd3 (Ochiai K. et al. JBC 2006; Tamahara T. et al. MCB 2017). Thus, the extracted 6,957 genes contained genes released from BACH2-mediated heterochromatin upon B cell activation.

We have modified the manuscript as below.

(Page 8, line 223)

“Importantly, 6,957 genes were extracted as genes which lost both BACH2 binding and H3K9me3 modification (Figure 2B, Table S2). **These genes included known BACH2 target genes related to plasma cell differentiation, such as *Prdm1* and *Ccnd3* (ref. ^{10, 19}). Thus, the extracted 6,957 genes contained genes released from BACH2-mediated heterochromatin upon B cell activation.**”

3) Fig. 3. Why do the authors pick up *Aicda* in naïve B cells? It is not clear how the authors are purifying the B cells from the mouse spleen, just looking at the data it is possible that the purification is not clean and there are activated B cells in this population. Is there any specific reason why they have used the B1-8 mouse for this particular analysis? It is also quite surprising that the authors can detect *Bcl6* expression in the naïve B cells. The authors should confirm this expression by repurifying bona fide naïve B cells (maybe via CD43-ve purification). Again it is unclear how they have purified the naïve B cells here.

REPLY:

[1] Why we picked up *Aicda* in naïve B cells (Figure 3A).

In Figure 1 and 2, we have explored BACH2-mediated heterochromatin formation of target gene loci. *Prdm1* and *Aicda* were released from heterochromatin upon BCR stimulation (Figure 2B). BACH2-mediated heterochromatin formation was supposed to be disturbed in *Bach2*-deficient naïve B cells, while transcripts of *Prdm1* but not *Aicda* were increased in these cells (Figure 3A). We speculate that IRF4 function is involved in the cause of difference: *Prdm1* is activated by IRF4 homodimer, while *Aicda* is activated by IRF4-BATF heterodimer which is transiently generated upon BCR stimulation (Ochiai K. Blood Advances 2018). In *Bach2*-deficient naïve B cells, accumulated IRF4 activated genes regulated by IRF4 homodimer motif (Figure 4A). On the other hand, the expression of BATF was not induced in *Bach2*-deficient naïve B

cells (Figure 3A), suggesting the failure of IRF4-BATF function. Although we have not focused on the function of IRF4-BATF in this study, the alteration is one of important issues to be analyzed in *Bach2*-deficient B cells.

We have modified the manuscript as below.

(Page 16, line 480)

“Then, the expression of *Aicda* is induced by BATF and a lower protein amount of IRF4 (ref. ²⁸), followed by the promotion of CSR. At that time, the expression of *Prdm1* and *Xbp1* is kept at a low level. However, their expression was induced by accumulated IRF4 in *Bach2*^{-/-} B cells (Figure 4B). **On the other hand, the expression of *Batf* and *Aicda* was not induced in *Bach2*-deficient naïve B cells (Figure 3A), suggesting the failure of IRF4-BATF function.**”

[2] Purification of naïve B cells from *Bach2*^{+/+} and *Bach2*^{-/-} mice spleen.

In this study, we purified splenic naïve B cells using B cell isolation kit from Miltenyi Biotec which deplete CD43-expressing B cells and non-B cells (<https://www.miltenyibiotec.com/US-en/products/b-cell-isolation-kit-mouse.html>). To present the feature of B cells sorted using the kit, we have provided flow cytometry data which shows the comparison of B cells before and after sorting (Supplemental data for Q3). About 25% of *Bach2*^{+/+} and *Bach2*^{-/-} splenic B cells were CD43-positive before sorting, and the majority of CD43-positive cells were depleted after sorting (the upper panels). These sorted *Bach2*^{+/+} or *Bach2*^{-/-} B cells were enriched with B220-positive cells (the intermediate panels). Naïve B cells express IgD and IgM on the cell surface, and activated B cells lose IgD expression. Sorted *Bach2*^{+/+} B cells express IgD (the lower panels), indicating unactivated B cells. These cells were utilized as *Bach2*^{+/+} naïve B cells. Compared with *Bach2*^{+/+} B cells, a portion of sorted *Bach2*^{-/-} B cells express lower IgD (the lower panel). *Bach2*^{-/-} B cells were already committed to PCs at transcript levels without antigen stimulation (Figure 3A). These cells were utilized as *Bach2*^{-/-} naïve B cells and investigated the feature in this study.

[3] Why we have used the B1-8 mice.

In previous reports, we have explored that BACH2 is regulated via the BCR-AKT-mTOR pathway (Tamahara T. et al. MCB 2017, Ando R. J Biol Chem 2016). However, it is not clear how BACH2 function is regulated under BCR signaling. To

explore the detail molecular mechanism of BACH2-mediated gene regulation under BCR signaling, we utilized B1-8 transgenic mice. B1-8 transgenic mice allow us to investigate B cell response against BCR using NP-ficol (Shih TY. et al. Nature Immunology 2002). We have established the *in vitro* BCR-mediated plasma cell differentiation system using B1-8 splenic naïve B cells (Ochiai K. et al. STAR Protocols 2021). Using the cell system, we investigated BACH2 function in this study. We have also generated *Bach2*-deficient mice with B1-8 background and investigated plasma cell differentiation in the absence of *Bach2* under BCR stimulation.

We have modified the manuscript as below.

(Page 6, line 151)

“In previous reports, we have set up an *in vitro* PC differentiation system using resting B (naïve B) cells purified from B1-8 mice, which allowed us to analyze both B cell activation and PC differentiation in response to BCR signaling³⁶ (Figure 1A). Considering that BACH2 is regulated via the BCR-AKT-mTOR pathway^{18, 19}, we examined how BACH2 function was regulated under BCR signaling using this system.”

(Page 9, line 252)

“We next generated *Bach2*-deficient mice with B1-8^{hi} background (hereafter *Bach2*^{-/-} mice), and performed transcriptome analysis comparing B cells purified from wild-type (*Bach2*^{+/+}) and *Bach2*^{-/-} mice, and examined transcripts of BACH2 target genes (Figure 3A).”

[4] Interpretation of detecting *Bcl6* expression in the naïve B cells.

There have been several reports showing BCL6 expression in naïve B cells. For example, BCL6 is expressed in naïve B cells (Muto A. et al. EMBO J 2010) and represses *Prdm1* with BACH2 cooperatively (Ochiai K. et al. Int Immunol. 2008). Of course, its expression is highly induced upon BCR stimulation (Ochiai K. et al. Immunity 2013). Considering that transcriptional function of PU.1 is activated by AKT (Rieske P. et al. J Biol Chem 2001), PU.1-IRF4 regulate the expression of *Bcl6* in both naïve B cells and activated B cells. In addition, *Bach2*-deficient mice lacked the germinal center formation upon immunization (Muto A. et al. Nature 2004). The reduced DNA binding of PU.1 presumably resulted in the downregulation of *Bcl6*, followed by the failure of GC formation in these mice.

To improve the manuscript, we have added a one paragraph to elaborate the above

description.

(Page 17, line 511)

“In a previous report, we have shown that *Bach2*-deficient mice lacked the germinal center formation upon immunization¹³. BCL6, a critical regulator of GC formation⁵³, is expressed in naïve B cells¹⁶ and represses *Prdm1* with BACH2 cooperatively¹⁴. The expression is highly induced upon BCR stimulation by PU.1-IRF4 (ref. ²⁴). Considering that transcriptional function of PU.1 is activated by AKT⁵⁴, PU.1-IRF4 regulate the expression of *Bcl6* in both naïve B cells and activated B cells. The reduced DNA binding of PU.1 presumably resulted in the downregulation of *Bcl6*, followed by the failure of GC formation in *Bach2*-deficient mice. Including *Bcl6* and *Pten*, the expression of EICE-regulated genes was reduced in FO B cells but not in MZ B cells in *Bach2*^{-/-} B cells (Figure 7B), indicating the reduced PU.1 functions in *Bach2*^{-/-} FO B cells. It will be necessary to understand the alteration of transcriptome as well as the PU.1 activity in *Bach2*^{-/-} FO B cells and MZ B cells.”

The following references have been added in the manuscript.

53. Basso, K. & Dalla-Favera, R. BCL6: master regulator of the germinal center reaction and key oncogene in B cell lymphomagenesis. *Advances in immunology* **105**, 193-210 (2010).
54. Rieske, P. & Pongubala, J.M. AKT induces transcriptional activity of PU.1 through phosphorylation-mediated modifications within its transactivation domain. *The Journal of biological chemistry* **276**, 8460-8468 (2001).

4) Fig. 7. Are the experiments done in B cells derived from immunized mice? If not, it is surprising to find Bcl6 expression (as mentioned above). What is the frequency of GCs in these spleen? Additionally, mb1-cre will delete Bach2 in early B cells. Is there a developmental defect? What is the rationale for using B1-8 mice?

REPLY:

[1] Interpretation of detecting *Bcl6* expression in B cells from unimmunized mice.

In this study, we used B cells purified from non-immunized mice. BCL6 is expressed in naïve B cells and highly expressed upon activation as described in Q3[4]. *Bach2*-deficient mice lacked the germinal center formation upon immunization (Muto A.

et al. Nature 2004).

As same as the reply to Q3[4], we have added the paragraph to the manuscript as below.
(Page 17, line 511)

“In a previous report, we have shown that *Bach2*-deficient mice lacked the germinal center formation upon immunization¹³. BCL6, a critical regulator of GC formation⁵³, is expressed in naïve B cells¹⁶ and represses *Prdm1* with BACH2 cooperatively¹⁴. The expression is highly induced upon BCR stimulation by PU.1-IRF4 (ref. ²⁴). Considering that transcriptional function of PU.1 is activated by AKT⁵⁴, PU.1-IRF4 regulate the expression of *Bcl6* in both naïve B cells and activated B cells. The reduced DNA binding of PU.1 presumably resulted in the downregulation of *Bcl6*, followed by the failure of GC formation in *Bach2*-deficient mice. Including *Bcl6* and *Pten*, the expression of EICE-regulated genes was reduced in FO B cells but not in MZ B cells in *Bach2*^{-/-} B cells (Figure 7B), indicating the reduced PU.1 functions in *Bach2*^{-/-} FO B cells. It will be necessary to understand the alteration of transcriptome as well as the PU.1 activity in *Bach2*^{-/-} FO B cells and MZ B cells.”

[2] About the rationale for using B1-8 mice.

BACH2 is regulated via the BCR-AKT-mTOR pathway (Tamahara T. et al. MCB 2017, Ando R. J Biol Chem 2016), and B1-8 mice are useful to analyze B cell response against BCR (Shih TY. et al. Nature Immunology 2002). In this study, we examined how BACH2 function was regulated under BCR signaling using the B1-8 B cell system. As same as the reply to Q3[3], we have modified the manuscript as below.

(Page 6, line 151)

“In previous reports, we have set up an *in vitro* PC differentiation system using resting B (naïve B) cells purified from B1-8 mice, which allowed us to analyze both B cell activation and PC differentiation in response to BCR signaling³⁶ (Figure 1A). Considering that BACH2 is regulated via the BCR-AKT-mTOR pathway^{18, 19}, we examined how BACH2 function was regulated under BCR signaling using this system.”

[3] B cell development and inflammation phenotypes in *Bach2*-deficient mice and Mb1-Cre:*Bach2*flox/flox mice (*Bach2* cKO).

We have shown the data comparing *Bach2*-deficient mice and *Bach2* cKO mice in Supplemental data for Q4A-C.

(A) Data presents the analysis of B cell development. *Bach2*-deficient mice showed the reduced cell numbers from pro-B to mature B cell. Compared with *Bach2*-deficient mice, *Bach2* cKO showed a normal cell number of pro-B and pre-B cell and the reduced cell number from immature B cells.

(B and C) Data shows the inflammation phenotypes. It should be noted that *Bach2*-deficient mice are exposed under severe systematic inflammation (Nakamura A. J Exp Med 2013), while *Bach2* cKO mice are not. For example, the lung of *Bach2*-deficient mouse was swelling for inflammation (B) (Nakamura A. J Exp Med 2013), and neutrophils were accumulated in the bone marrow and the spleen (C (1)). These manifestations of inflammation were not observed in *Bach2* cKO (B and C (2)). Thus, the phenotype of B cell development is different between *Bach2*-deficient mice and *Bach2* cKO mice. The inflammation circumstance is supposed to affect early B cell development in *Bach2*-deficient mice which showed abnormal functions of not only B cells but also other immune cells, as discussed in the manuscript (page 17, line 523). In addition, these observations in *Bach2* cKO mice also indicate that BACH2 becomes essential for B cells after expressing BCR. Since this manuscript already reports diverse data sets, we would like investigate how *Bach2*-deficiency influences on B cell function after expressing BCR as separate, future projects.

Referee #3:

In this manuscript, the authors investigated how the roles of BACH2 and IRF4 were integrated into the regulation of CSR and PC differentiation. The authors reported that BACH2 organized heterochromatin formation of target genes including IRF4 target genes and PRDM1. They reported that PU.1-IRF4 promotes BACH2 function by inducing BACH2 expression and reducing AKT activity. They demonstrated that BACH2 and IRF4 mutually modulate their biological functions and that BACH2 inhibits PC differentiation repressing PC genes and the restriction IRF4 protein accumulation.

The paper is well written with a large data set of experiments. Data analysis has been thoroughly performed and support the conclusion. Supplemental data ideally complement main figures.

I can come with some hopefully constructive comments:

REPLY: We appreciate this reviewer for many suggestions. We have modified the manuscript according to comments during the course of revision, specific points are addressed as below. Since this manuscript already reports diverse data sets, we would like to carry out some of the suggested experiment like single cell transcriptome as separate, future projects.

1) The heme was shown to directly bind and inhibit BACH2 function (Watanabe-Matsui M et al. Blood 2011; Jang KJ et al. Nat Comm 2015). Is there a role of heme in the presented results?

REPLY: Thank you for your important comment. Along this line, we have been working on BACH2 protein interactome which is regulated by heme (Watanabe-Matsui M. et al. Blood 2011). We have some interesting candidates but would like to consider the role of heme in the regulation between BACH2 and IRF4 as one of following issues.

We have added the description in the manuscript as below.

(Page 18, line 542)

“BACH2 is a critical regulator for the acquisition of antigen-specific antibodies, and such BACH2 function is regulated by cellular molecules including mTORC1 and heme^{18, 19, 65}.”

The following reference has been added in the manuscript.

65. Watanabe-Matsui, M. *et al.* Heme regulates B-cell differentiation, antibody class switch, and heme oxygenase-1 expression in B cells as a ligand of Bach2. *Blood* **117**, 5438-5448 (2011).

2) Inhibition of Polycomb PRC2 complex was shown to accelerate PC differentiation (Herviou L et al. Leukemia 2019). Did the authors investigated the involvement of H3K27me3 in BACH2-mediated target gene repression?

REPLY: This is an important possibility we were also interested in. However, BACH2 complex in primary B cells did not contain major components of the PRC2 complex. Therefore, we have not examined further the involvement of PRC2 complex. While we cannot exclude the involvement of H3K27me3, we like to focus on the importance of H3K9me3-related factors in BACH2-mediated target gene repression in this manuscript.

We have added the description in the manuscript as below.

(Page 15, line 459)

“The BACH2 complexes purified from primary B cells contained the H3K9me3-mediated heterochromatin factors, such as TRIM28, HP1 γ and histone H1, as well as the nuclear membrane protein LAMIN B1 (Figure 1E). BACH2 was co-localized with TRIM28 or HP1 γ in the proximity of the nuclear membrane in naïve B cells (Figure 1G, H). In our analysis, the BACH2 complexes did not contain major components of the PRC2 complex which promotes the formation of H3K27me3-mediated heterochromatin⁴⁵. Enhancer of zest 2 (EZH2), the catalytic subunit of the PRC2 complex, is highly induced in GC B cells and represses plasma cell differentiation^{46, 47, 48}. While we cannot exclude the involvement of H3K27me3 in BACH2-mediated gene regulation, it was not the case in the cells we examined in this

study.”

The following references have been added in the manuscript.

45. Wiles, E.T. & Selker, E.U. H3K27 methylation: a promiscuous repressive chromatin mark. *Current opinion in genetics & development* **43**, 31-37 (2017).
46. Velichutina, I. *et al.* EZH2-mediated epigenetic silencing in germinal center B cells contributes to proliferation and lymphomagenesis. *Blood* **116**, 5247-5255 (2010).
47. Béguelin, W. *et al.* EZH2 and BCL6 Cooperate to Assemble CBX8-BCOR Complex to Repress Bivalent Promoters, Mediate Germinal Center Formation and Lymphomagenesis. *Cancer cell* **30**, 197-213 (2016).
48. Herviou, L., Jourdan, M., Martinez, A.M., Cavalli, G. & Moreaux, J. EZH2 is overexpressed in transitional preplasmablasts and is involved in human plasma cell differentiation. *Leukemia* **33**, 2047-2060 (2019).

3) Early UPR activation mediated by mTORC1 was reported in activated B cells (Gaudette BT et al. Nat Comm 2020) driving PC priming before XBP1 gene expression. The authors could investigate how it could be affected in BACH2^{-/-} cells.

REPLY: It is an important suggestion. Since mTORC1 was activated in *Bach2*-deficient B cells, we also assume the possibility of early UPR activation in these cells. Especially, we are interested in whether early UPR activation affects IRF4 regulation in *Bach2*-deficient cells. However, for the sake of clarity of the current manuscript, we would like to investigate these issues as a next project.

4) Regarding the figure 7, it would be interesting to have a complete transcriptome analysis of Bach2^{-/-} FO and MZ B cells.

REPLY: In Figure 7A and B, we have compared the expression of selected genes regulated by PU.1-IRF4 or IRF4-IRF4, and confirmed the regulation of IRF4 function by BACH2 in FO B cells. This result has also suggested the importance to identify gene

sets altered in *Bach2*-deficient FO and MZ B cells. Transcriptome analysis would reveal the overall picture, and extend our understanding about how *Bach2*-deficiency affect the expression of genes in FO and MZ B cells. We consider to perform it as a following analysis. Thank you for the important suggestion.

Considering the importance of PU.1-IR4 function in FO B cells (Figure 7B), we have added a one paragraph to improve the manuscript.

(Page 17, line 511)

“In a previous report, we have shown that *Bach2*-deficient mice lacked the germinal center formation upon immunization¹³. BCL6, a critical regulator of GC formation⁵³, is expressed in naïve B cells¹⁶ and represses *Prdm1* with BACH2 cooperatively¹⁴. The expression is highly induced upon BCR stimulation by PU.1-IRF4 (ref. ²⁴). Considering that transcriptional function of PU.1 is activated by AKT⁵⁴, PU.1-IRF4 regulate the expression of *Bcl6* in both naïve B cells and activated B cells. The reduced DNA binding of PU.1 presumably resulted in the downregulation of *Bcl6*, followed by the failure of GC formation in *Bach2*-deficient mice. Including *Bcl6* and *Pten*, the expression of EICE-regulated genes was reduced in FO B cells but not in MZ B cells in *Bach2*^{-/-} B cells (Figure 7B), indicating the reduced PU.1 functions in *Bach2*^{-/-} FO B cells. It will be necessary to understand the alteration of transcriptome as well as the PU.1 activity in *Bach2*^{-/-} FO B cells and MZ B cells.”

The following references have been added in the manuscript.

53. Basso, K. & Dalla-Favera, R. BCL6: master regulator of the germinal center reaction and key oncogene in B cell lymphomagenesis. *Advances in immunology* **105**, 193-210 (2010).
54. Rieske, P. & Pongubala, J.M. AKT induces transcriptional activity of PU.1 through phosphorylation-mediated modifications within its transactivation domain. *The Journal of biological chemistry* **276**, 8460-8468 (2001).

5) The analysis of PC differentiation comparing BACH2^{-/-} and BACH2^{+/+} B cells at the single cell level could be important to understand how the maturation trajectories are affected.

REPLY: Taking into account the severe systemic inflammation in *Bach2*-deficient mice,

the single cell level analysis will reveal the maturation trajectories under inflammation. As well as the transcriptome analysis of FO and MZ B cells, we consider it as one of prioritized experiments to be done in future.

6) The authors could discuss more the results presented in Figures 5 C, D and E demonstrating that results obtained with the BACH2^{-/-} B cells are not identified in B cell specific Bach2 depletion model.

REPLY: To improve the pointed issue, we have added RT-PCR data which examined the expression of EICE-regulated genes *Bcl6*, *Icosl* and *Pten* in control and *Bach2* cKO B cells (Revised Figure S5A). Data showed that the expression of these genes was not severely altered in *Bach2* cKO B cells, supporting the unchanged pAKT level and PU.1-IRF4 function in *Bach2* cKO B cells. In addition, it emphasizes the abnormality occurred in *Bach2*-deficient B cells.

We have added the description of the above data in the manuscript as below.

(Page 14, line 419)

“We found that the above aberrant PC differentiation observed in *Bach2*^{-/-} mice was not the case in B-cell specific *Bach2*-deficient (Mb1-Cre:*Bach2*^{fl/fl}, hereafter cKO) mice. In *Bach2* cKO B cells, IRF4 protein was not accumulated (Figure 7C), and AKT was not activated, accompanied with the no altered expression of EICE-regulated genes, *Bcl6*, *Icosl* and *Pten* (Figure 7D, S5A).”

Minor comments:

7) P16: typo error.

REPLY: Thank you for pointing out our error. We have removed typographical error, “阻止”, from the manuscript. (Page 17, line 531)

Figure for reviewers removed

Supplemental data for Q3.
Splenic naive B cells purified using the B cell isolation kit from Miltenyi Biotec.

Figure for reviewers removed

Dear Prof. Ochiai,

Congratulations on a great revision! Overall, the referees are in support of publication. However there remain several editorial items that we ask you to attend to in a revised version.

When you submit your revised version, please include a point-by-point response which addresses the following:

1. Please add the funding 20KO7321 into EJP online.
2. Please remove the author contribution section from the main manuscript.
3. Please review our policy on conflicts of interests and update the title of this section to: "Disclosure and competing interests statement"
4. Please move the references before the figure legends and correct to be listed in alphabetical order. Also, please ensure only ten authors are listed, followed by et al.
5. We include a synopsis of the paper (see <http://emboj.embojournal.org/>). Please provide me with a general summary statement and 3-5 bullet points that capture the key findings of the paper. 12.
6. We also need a summary figure for the synopsis. The size should be 550 wide by 200-440 high (pixels). You can also use something from the figures if that is easier.
7. Table S1 and S6 should be renamed "Appendix Table S1" and "Appendix Table S2" and be added to the appendix file, with their legends. Table S2-5 should be uploaded in excel format as individual files for better readability, and renamed "Dataset EV1" - "Dataset EV3". Legends should be added to the corresponding file as well, on a separate worksheet.
8. Please move the material and methods after the discussion.
9. Please note that a separate 'Data Information' section is required in the legends of figures 1. 1b, g-h; 3b, d-e; 5i-j, l-o, supplementary figures 5a-c.
10. Please indicate the statistical test used for data analysis in the legends of figures 2e, h; 3a; 4d; 5a, c, 7e, supplementary figures 1; 5a-b.
11. Please note that the error bars are not defined in the legends of figure 7e, supplementary figures 5a-b.
12. Please note that the measure of center for the error bars needs to be defined in the legends of figures 3d; 4h; 5i, l, n; 7b.
13. Please note that the red arrowhead is not defined in the legend of figure 6f. This needs to be rectified.
14. The link provided for PRJNA998173 does not work, please rectify.

Thank you for the opportunity to consider your work for publication, I look forward to the revised version.

Kind regards,

Kelly

Kelly M Anderson, PhD
Editor, The EMBO Journal
k.anderson@embojournal.org

Further information is available in our Guide For Authors: <https://www.embojournal.org/page/journal/14602075/>

authorguide

Referee #2:

The authors have satisfactorily addressed the points raised by this reviewer. The revised manuscript is a much better version of the original submission and is now ready for publication.

Referee #3:

The authors provided answers to all of my comments.
The manuscript is now suitable for publication.

The authors addressed the remaining editorial issues.

Dear Prof. Ochiai,

Congratulations on an excellent manuscript, I am pleased to inform you that your manuscript has been accepted for publication in the EMBO Journal. Thank you for your comprehensive response to the referee concerns and for providing detailed source data. It has been a pleasure to work with you to get this to the acceptance stage.

I will begin the final checks on your manuscript before submitting to the publisher next week. Once at the publisher, it will take about 3 weeks for your manuscript to be published online. As a reminder, the entire review process, including referee concerns and your point-by-point response, will be available to readers.

I will be in touch throughout the final editorial process until publication. In the meantime, I hope you find time to celebrate!

Warm wishes,
Kelly

Kelly M Anderson, PhD
Editor, The EMBO Journal
k.anderson@embojournal.org
